

# Ensemble Forecasts of Air Quality in Eastern China

## Part 1. Model Description and Implementation
## of the MarcoPolo-Panda Prediction System

Guy P. Brasseur
Max Planck Institute for Meteorology, Hamburg, Germany
and National Center for Atmospheric Research, Boulder, CO, USA

Ying Xie
Shanghai Meteorological Service, Shanghai, China

A. Katinka Petersen
Max Planck Institute for Meteorology, Hamburg, Germany

Idir Bouarar
Max Planck Institute for Meteorology, Hamburg, Germany

Johannes Flemming
European Centre for Middle Range Weather Forecasts, Reading, UK.

Michael Gauss
Norwegian Meteorological Institute, Oslo, Norway

Fei Jiang
Nanjing University, Nanjing, China

Rostislav Kouznetsov
Finnish Meteorological Institute, Helsinki, Finland.

Richard Kranenburg
TNO, Utrecht, The Netherlands

Bas Mijling
Royal Netherlands Meteorological Institute (KNMI), De Bilt, The Netherlands

Vincent-Henri Peuch
European Centre for Middle Range Weather Forecasts, Reading, UK

Matthieu Pommier
Norwegian Meteorological Institute, Oslo, Norway

Arjo Segers
TNO, Utrecht, The Netherlands

Mikhail Sofiev
Finnish Meteorological Institute, Helsinki, Finland

Renske Timmermans
TNO, Utrecht, The Netherlands

Ronald van der A
Royal Netherlands Meteorological Institute (KNMI), De Bilt, The Netherlands,
and Nanjing University of Information Science and Technology, Nanjing, China

Stacy Walters
National Center for Atmospheric Research, Boulder, CO, USA

Jianming Xu
Shanghai Meteorological Service, Shanghai, China

Guangqiang Zhou
Shanghai Meteorological Service, Shanghai, China



## Abstract

An operational multi-model forecasting system for air quality including 9 different chemical transport models has been developed and is providing daily forecasts of ozone, nitrogen oxides, and particulate matter for the 37 largest urban areas of China (population higher than 3 million in 2010). These individual forecasts as well as the mean and median concentrations for the next 3 days are displayed on a publicly accessible web site (www.marcopolo-panda.eu). The paper describes the forecasting system and shows some selected illustrative examples of air quality predictions. It presents an inter-comparison of the different forecasts performed during a given period of time (1-15 March 2017), and highlights recurrent differences between the model output as well as systematic biases that appear in the median concentration values. Pathways to improve the forecasts by the multi-model system are suggested.





## 1. Introduction

The rapid economic growth in China has been accompanied with a substantial degradation
of air quality, particularly in the densely populated areas of the eastern part of the country.
Air pollution is the source of cardiovascular and respiratory illness, increased stress to heart
and lungs and cell damage in the respiratory system, which in turn can result in fatalities
resulting from ischemic heart disease, chronic obstructive pulmonary disease (COPD) and
Lower Respiratory Infections. To address this problem, China is taking effective measures to
reduce the emission of primary pollutants such as nitrogen oxides (NOx), volatile organic
compounds (VOCs) and particulate matter (PM). In addition to these long-term mitigation
measures, immediate action can be taken to avoid the occasional occurrence of acute air
pollution episodes, particularly in winter during stable meteorological situations, by
drastically reducing emissions associated with polluting activities during the periods of
predicted events. The implementation of such measures requires that accurate forecasts of
air quality be produced and made available to local and regional authorities. Alerts to warn
the public of the imminence of acute pollution episodes can be released several days before
the event on the basis of model predictions.
Advanced forecast models include a detailed formulation of the chemical and physical
processes responsible for the formation of secondary pollutants such as ozone and
particulate matter in response to the emissions of primary species produced as a result of
industrial, agricultural and residential activities, energy production and transportation.
These models simulate the transport of these constituents by the atmospheric circulation as
well as vertical exchanges by convective motions and turbulent boundary layer mixing.
Meteorological information provided by weather forecast models is therefore an essential
input to regional air quality models. Surface deposition of oxidized compounds and wet
scavenging of soluble species are also taken into account. The atmospheric concentrations
of the chemical and physically interacting species are obtained by solving a mathematically
stiff system of partial differential equations with appropriate initial and boundary
conditions.
The approach used to produce predictions of air quality bears a lot of resemblance with the
methods used for weather forecasts. In both cases, models make use of similar numerical
algorithms, include assimilated data, produce large amounts of output that have to be
analysed and evaluated, and eventually disseminated to the public in the form of easily
accessible information. The steady progress made in the numerical weather prediction since
the 1980's (Bauer et al., 2015), through combined scientific, computational and
observational advances, has also considerably improved our capability of providing
predictive information on air quality and on its impacts for human society (i.e., health, food
production and the state of ecosystems).
Many models are available for operationally forecasting air quality [Kukkonen et al., 2012]
and have been tested in different contexts. These models are usually driven by different
input data (surface emissions, weather forecasts, chemical schemes, aerosol formulation,
land use data, boundary conditions, etc.) and hence generate different output (e.g.,
different concentrations of chemical species). In most cases, it is difficult to clearly



distinguish between models that perform well and models that perform poorly because the
success of individual models varies with the conditions that are encountered (e.g.,
geographic location, season, meteorological situation) and can be different for the different
chemical species and for different statistical parameters. If the models involved have been
developed rather independently from each other their results can be combined and their
individual behaviours can be examined by comparing the predicted fields to the median or
the mean derived from the ensemble of simulations. Much can be learned from a
systematic day-by-day examination of the model behaviour operated in a forecast mode.
Building ensemble of models is an attractive approach to forecast air quality, because the
inter-model variability provides insight on the robustness of the results or conversely on
their uncertainties [Vautard et al., 2006; Solazzo et al., 2012]. Further, the composite
products have usually better overall performance than the results produced by individual
systems [Galmarini et al., 2013; Riccio et al., 2007; Sofiev, 2015; 2017]. This approach is
especially useful in the context of decision-making since it samples the uncertainty space
associated with the different individual forecasts.
Numerical weather and seasonal forecasting are usually based on a single model ensemble
in which the initial conditions are slightly perturbed so that different likely evolutions of the
atmospheric dynamics can be projected. In the case of air quality forecasts, which are not
only initial value problems, it is advisable to also perturb emissions, meteorology and
boundary conditions as well as model parameters (kinetic reaction rates…), which is best
performed by considering a multi-model ensemble [Dabberdt and Miller, 2000].
Nevertheless, in addition, it would also be useful to assess the behaviour of a single air
quality model that is driven by different realizations of ensemble meteorological forecasts.
The models used in the present study have been developed rather independently, and this
leads to a rather broad range of model results. Model performance does not only depend
on the quality of emissions datasets: they differ for a wide range of reasons, including
dynamical and weather aspects but also the adopted formulation (parameterisations,
operator splitting, time integration…) and numerical algorithms. An inspection of the
different choices made in the models can lead to some improvements in model
configurations, and hence will reduce the "artificial" spread between calculated fields. This
spread often results from errors in the configuration (set-up bugs…) or from inaccuracies in
the adopted input parameters (land-use…). By including each model configuration within a
large ensemble, the combined performance of the forecast system is considerably less
affected by initial implementation issues or inadequate choice of input parameters applied
in individual models.
This paper describes the early phase of a system that forecasts air quality in eastern China.
The system can be characterized as a multi-model "ensemble of opportunity" (as defined by
a combination of models running in their default configurations) that is evolving into an
operational air quality ensemble prediction system, similar to the system established in
Europe under the Copernicus Atmospheric Monitoring Service (CAMS) [Marecal et al.,
2015]. The concept adopted here will be briefly presented in Section 2. Section 3 presents a
description of the different models and Section 4 discusses the performance of the whole



system and of the contributing models. Approaches to improve the performance of the
system are presented in Section 5.
The ensemble of models considered in the present study has been assembled under the
Panda and MarcoPolo projects supported by the European Commission within the
Framework Programme 7 (FP7). Seven models were initially included in the operational
system: the global IFS model developed and operated by the European Centre for Middle
Range Weather Forecasts (ECMWF), five regional models implemented by European
research and service institutions (CHIMERE by the Royal Netherlands Meteorological
Institute (KNMI), WRF-Chem-MPIM by the Max Planck Institute for Meteorology (MPIM),
SILAM by Finnish Meteorological Institute (FMI), EMEP/MSC-W by the Norwegian
Meteorological Institute (MET.Norway), LOTOS-EUROS by The Netherlands Organisation for
Applied Scientific Research (TNO)), and one model (WRF-Chem-SMS) applied in China by the
Shanghai Meteorological Service (SMS). In later steps, forecasts by additional regional
models applied by Nanjing University (WRF-CMAQ) and by the Shanghai Meteorological
Service (WARMS-CMAQ) were added to the ensemble. In the following Section, we provide
a brief overview of these different models. Only seven of them contribute to the inter-
comparison presented in Section 4.
**2. Description of the Models included in the Ensemble**
In the following subsections, each of the 9 participating models will be described. Table 2a-b
presents the key characteristics of each model involved in the inter-comparison and Table 3
summarizes the emissions adopted in each model.
***2.1. IFS***
IFS (Integrated Forecasting System) is ECMWF's global Numerical Weather Prediction
system. As part of the past series of European projects MACC and now of CAMS, the
Copernicus Atmosphere Monitoring Service, IFS has been developed to represent optionally
chemical processes in the troposphere and in the stratosphere. Flemming et al. (2015)
provide a detailed description of the modelling of chemical processes in the IFS, and Inness
et al. (2015) describe the aata assimilation aspects.
For the work presented here, the version of IFS used is Cycle 43R1 (see documentation at
https://www.ecmwf.int/en/forecasts/documentation-and-support/changes-ecmwf-
model/ifs-documentation). The model is run globally at a resolution of T511 (about 40km)
on the horizontal, and with 60 levels on the vertical extending up to the top of the
stratosphere. The chemical package used originates from the TM5 Chemistry and Transport
Model (Huijnen et al., 2010). It has been fully integrated into the IFS code and comprises 54
tracers and 120 reactions focusing on tropospheric ozone-CO-NMVOC chemistry. In the
configuration used here, stratospheric ozone is modelled with a simple linearized scheme.
Aerosols are represented using the scheme described by Morcrette et al. (2009), which
includes 5 species: dust, sea-salt, black carbon, organic carbon and sulphates. Tracers are
transported using the semi-Lagrangian scheme available in IFS with a mass fixer activated in
order to minimise mass non-conservation.



During the study period, IFS has been run twice daily (5-day forecasts) assimilating a range of satellite chemical data on top of the full list of meteorological satellite and non-satellite data that ECMWF uses for its medium-range weather forecasts. Table 1 indicates the satellite data streams actively assimilated for the experiments presented here. As a result, IFS forecasts benefit from all these observations to afford a realistic representation of large scales for weather parameters as well as, to some extent, for chemical variables (species assimilated).

IFS used the MACCITY emission data set updated for the year 2017. Biogenic emissions of VOC were taken from a climatology of a multi-year MEGAN model simulation. Daily emissions from biomass burning were derived from satellite retrieval of fire radiative power (FRP) from the MODIS instruments by the Global Fire Assimilation System (GFAS, Kaiser et al. 2012). The observed fire emissions from the day before the forecast start are used for all five days of the forecast. Desert dust and sea salt emissions were simulated online for each time step based on the IFS meteorological fields and the land use.

As part of CAMS, the chemical configuration of IFS benefits from routine detailed evaluations. Validation reports are produced quarterly and can be found here (http://atmosphere.copernicus.eu/quarterly_validation_reports). The report for the period March-May 2017 provides insight on the overall performance of the runs that are also presented here. Further information about the IFS code can be obtained from Vincent Henri Peuch          vincent-henri.peuch@ecmwf.int          and          on          the          web          site https://www.ecmwf.int/en/about/what-we-do/environmental-services/copernicus-atmosphere-monitoring-service

**Table 1. Satellite data streams (atmospheric composition variables only) assimilated in IFS.**

| Instrument | Satellite | Space Agency | Data Provider | Species |
|---|---|---|---|---|
| **MODIS** | EOS-Aqua, EOS-Terra | NASA | NASA | AOD |
| **MLS** | EOS-Aura | NASA | | O3 profile |
| **OMI** | EOS-Aura | NASA | KNMI | O3, NO2, SO2 |
| **SBUV-2** | NOAA-19 | NOAA | NOAA | O3 profile |
| **IASI** | METOP-A, METOP-B | EUMETSAT/CNES | ULB/LATMOS | CO |
| **MOPITT** | EOS-Terra | NASA | NCAR | CO |
| **GOME-2** | METOP-A, METOP-B | EUMETSAT/ESA | AC-SAF | O3, SO2 |
| **OMPS** | Suomi-NPP | NOAA | EUMETSAT | O3 |
| **PMAp** | METOP-A, METOP-B | EUMETSAT | EUMETSAT | AOD |

### 2.2. CHIMERE

CHIMERE is a regional chemistry-transport model used for analysis, scenarios and forecast (Menut et al., 2013). When used in the forecast mode, the model provides local scale information (to be compared with data from numerous air quality networks), or regional scale information (e.g., the French PREVAIR and the Copernicus CAMS systems). CHIMERE is





an open-source model, freely distributed at www.lmd.polytechnique.fr/chimere. In this
version, CHIMERE is used in off-line mode, forced by pre-calculated hourly meteorological
fields for the dynamics and by several emissions fluxes for the chemistry. The emissions are
pre-calculated or on-line estimated in the model with anthropogenic emissions (MEIC 2010),
biogenic emissions with the online model of emissions of gases and aerosols from nature
(MEGAN, Guenther et al., 2006), mineral dust (Menut et al., 2013) and biomass burning
emissions (Turquety et al., 2014). The gas phase chemistry is calculated using the
MELCHIOR2 mechanism and the aerosols are represented using a distribution of 10 bins,
from 40nm to 40μm to well describe both number and mass. The chemical boundary
conditions are provided by the LMDz-INCA model for gas and particles (Szopa et al., 2009),
except for mineral dust extracted from global GOCART simulations (Ginoux et al., 2001).
Further information about the implementation of the model for air quality forecasts in
China can be obtained from Ronald van der A (avander@knmi.nl) at KNMI and on the web
site http://www.lmd.polytechnique.fr/chimere/CW-download.php .
***2.3. WRF-Chem-MPIM***
The Weather Research and Forecasting model coupled to chemistry (WRF-Chem) is a
mesoscale non-hydrostatic meteorological model (Skamarock et al., 2008) coupled "online"
with chemistry that simultaneously predicts meteorological and chemical components of
the atmosphere (Grell et al., 2005; Fast et a., 2006).
WRF-Chem-MPIM is based on version 3.6.1 of the WRF-Chem model coupled to the gas
phase chemistry and the aerosol microphysics schemes provided by the Model for Ozone
and Related Chemical Tracers (MOZART-4, Emmons et al., 2010) and the Model for
Simulating Aerosol Interactions and Chemistry (MOSAIC, Zaveri et al., 2008), respectively.
Aerosols sizes are represented by four consecutive bins, and the formation of secondary
organic aerosol (SOA) from anthropogenic precursors is parameterized according to Hodzic
and Jimenez (2011).
Two nested model domains with horizontal resolutions of 60 km (Asian continent from India
to Japan) and 20 km (eastern China), respectively are implemented. The vertical grid is
composed of 51 levels extending from the surface to 10 hPa (~30 km).  A more complete
description of the selected physical and chemical options is provided in the WRF and in the
WRF-Chem user's guides under
http://www2.mmm.ucar.edu/wrf/users/docs/user_guide_V3.6/ARWUsersGuideV3.6.1.pdf
and https://ruc.noaa.gov/wrf/wrf-chem/Users_guide.pdf.
The WRF-Chem-MPIM model forecasts are initialized and forced at the lateral boundaries
every day by 6 hourly meteorological analysis data from the NCEP Global Forecast System
(GFS) at 0.5 degree resolution. For the chemical and aerosol species, 6 hourly datasets are
provided by the global operational forecasting system implemented within the Copernicus
Atmospheric Monitoring Service project (Flemming et al., 2015). More information on the
model's configuration can be obtained from Idir Bouarar (idir.bouarar@mpimet.mpg.de) at
the Max Planck Institute for Meteorology and on the web site
http://www2.mmm.ucar.edu/wrf/users/downloads.html.





### 2.4. SILAM

FMI uses the SILAM model version 5.5 (Sofiev et al., 2015). SILAM includes a meteorological pre-processor for diagnosing the basic features of the boundary layer and the free troposphere from the meteorological fields provided by various meteorological models (Sofiev et al., 2010). The dry deposition scheme for particles is described in Kouznetsov and Sofiev (2012). The surface resistance model for gases is based on a modified Wesely scheme (Wesely, 1989).

The gas phase chemistry was simulated with CBM-IV, with reaction rates updated according to the recommendations of IUPAC (http://iupac.pole-ether.fr) and JPL (http://jpldataeval.jpl.nasa.gov) and the terpenes oxidation added from CB05 reaction list (Yarwood et al., 2005). The sulphur chemistry and secondary inorganic aerosol formation is computed with an updated version of the DMAT scheme (Sofiev, 2000) and secondary organic aerosol formation with the Volatility Basis Set (VBS, Donahue et al., 2006), the volatility distribution of anthropogenic OC taken from Shrivastava et al. (2011).

The MACCITY land-based emissions are used together with STEAM shipping emissions. The simulations include sea-salt emissions as in Sofiev et al. (2011), biogenic VOC (volatile organic compounds) emissions as in Poupkou et al. (2010) and wild-land fire emissions as in Soares et al. (2015) and desert dust.

The grid cell size was roughly 15km × 10km ($0.125° × 0.125°$) covering the whole China, India, Japan and several countries of South-East Asia (67E, 7N) – (147E, 54N). The Asian forecasts are nested into the SILAM global AQ forecasts (http://silam.fmi.fi), from where they take lateral and top boundary conditions. The initial conditions for each run are taken from the previous-day forecast or, in case of failure, from global computations. Detailed information about the SILAM modelling system can be obtained from Mikhail Sofiev (Mikhail.Sofiev@fmi.fi) and from Rostislav Kouznetsov (rostislav.kouznetsov@fmi.fi) and on the web site of the Finnish Meteorological Institute (http://silam.fmi.fi/).

### 2.5. EMEP

The EMEP/MSC-W model (hereafter referred to as 'EMEP model') is a 3-D Eulerian Chemical Transport Model described in detail in Simpson et al. (2012). Although the model has traditionally been aimed at European simulations, global modelling has been possible for many years (Jonson et al., 2010; Wild et al., 2012). The EMEP configuration for the present study covers the East-Asian domain [15°N-55°N] x [90°E-135°E] with a horizontal resolution of 0.1° x 0.1° (longitude-latitude). The model uses 20 vertical levels defined as sigma coordinates. The 10 lowest levels are within the PBL, and the top of the model domain is at 100 hPa.

Particulate (PM) emissions are split into elementary carbon (EC), organic matter (OM) (here assumed inert) and the remainder, for both fine and coarse PM. The OM emissions are further divided into fossil fuel and wood-burning compounds for each source sector. As in Bergström et al. (2012), the Organic Matter/Organic Carbon ratio of emissions by mass is assumed to be 1.3 for fossil-fuel sources and 1.7 for wood-burning sources. The model also





calculates windblown dust emissions from soil erosion. Secondary PM2.5 aerosol consists of
inorganic sulphate, nitrate and ammonium, and SOA; the latter is generated from both
anthropogenic and biogenic emissions (anthropogenic SOA and biogenic SOA respectively),
using the 'VBS' scheme detailed in Bergström et al (2012) and Simpson et al (2012).
Model updates since Simpson et al. (2012), resulting in EMEP model version rv4.9 as used
here, have been described in Simpson et al. (2016) and references cited therein. The main
changes concern a new calculation of aerosol surface area, revised parameterizations of
$N_2O_5$ hydrolysis on aerosols, additional gas-aerosol loss processes for $O_3$, $HNO_3$ and $HO_2$, a
new scheme for ship $NO_x$ emissions, and the use of new maps for global leaf-area (used to
calculate biogenic VOC emissions) – see Simpson et al. (2015) for details. The EMEP model,
including a user guide, is publicly available as Open Source code at
https://github.com/metno/emep-ctm. For more details, please contact Michael Gauss
(michael.gauss@met.no).
The EMEP forecasts are driven by 3-hourly meteorological forecast data from the ECMWF
IFS model at 0.1 degree resolution. As for WRF-Chem, 6-hourly datasets for the chemical
and aerosol species are provided by the global operational forecasting system implemented
within the Copernicus Atmospheric Monitoring Service project.

### 369 *2.6. LOTOS-EUROS*

LOTOS-EUROS is a three-dimensional regional chemistry transport model (CTM) for
simulation of trace gases and aerosol concentrations in the boundary layer. Meteorological
input is obtained from an offline model, in this study from ECMWF. The model is of
intermediate complexity allowing long-term model simulations. For a detailed model
description we refer to Manders et al. (2017) and references therein.
In this study LOTOS-EUROS version 1.10 was used to simulate air quality over China. The
configuration is described by Timmermans et al. (2017) who adopted this version of the
model to investigate the origin of fine particulate matter across China using a source
apportionment technique. Through a one-way nesting procedure a simulation over East-
China was performed on a resolution of 0.25° longitude by 0.125° latitude, approximately 21
by 15 $km^2$. This domain is nested in a larger domain covering China almost entirely with a
resolution 1° longitude by 0.5° latitude, approximately 84 by 56 $km^2$. Chemical boundary
conditions for the coarse resolution domain were taken from the CAMS global modelling
framework (Flemming et al., 2015) and include trace gasses and aerosols. In the vertical, the
model used a boundary layer approach with 5 layers: a surface layer of 25m, a well-mixed
boundary layer, two reservoir layers, and a layer for the free troposphere. The boundary
layer height therefore defines the vertical structure of the model, and is here taken from
the meteorological input. More details about the code can be obtained by contacting
Renske Timmermans (renske.timmermans@tno.nl) at TNO or by consulting the web site
https://lotos-euros.tno.nl/.

### 393 *2.7. WRF-Chem-SMS*



WRF-Chem-SMS is based on WRF-Chem (Grell et al., 2005) version 3.2. The Regional Acid Deposition Model version 2 (RADM2, Chang et al., 1989) is used to represent gas-phase chemistry. ISORROPIA II is implemented to treat thermodynamic equilibrium for inorganic aerosols (Fountoukis and Nenes, 2007), and the Secondary ORGanic Aerosol Model (SORGAM) (Schell et al., 2001) is used to parameterize secondary organic aerosol formation. Madronich TUV scheme is applied for photolysis (Madronich and Flocke, 1999; Tie et al., 2003). The model domain covers the eastern region of China with horizontal resolutions of 6 km and 28 vertical layers. Biogenic emissions are calculated online using MEGAN model (Guenther et al., 2012). The multi-resolution emission inventory for China (MEIC inventory, http://www.meicmodel.org/; Li et al., 2014; Liu et al., 2015) for year 2010 is used to represent anthropogenic emissions.

The modeling system is initialized and forced at the lateral boundaries every day by 6 hourly data from the NCEP GFS at 0.5-degree resolution. For chemical species, previous modeling result is used for initial conditions. MOZART-4 historic data are employed as the gaseous chemical lateral boundary, and real time forecast of dust from the WRF-Dust model is employed as dust lateral boundary every 6 hours. More detailed information can be found in Zhou et al. (2017) and by contacting Jianming Xu (metxujm@163.com) at the Shanghai Meteorological Service.

### *2.8. WRF-CMAQ*

A regional air quality operational forecasting system was developed at Nanjing University, China, on the basis of the WRF-CMAQ model. The version adopted for the WRF and CMAQ models are V3.5 and V4.7.1, respectively. Two nested domains with horizontal resolutions of 36 km and 12 km are adopted for the forecasts. The outer domain covers the entire continental region of China as well as surrounding countries in East Asia. The inner domain mainly focuses on the densely populated area of eastern China. The number of grid points adopted for the WRF model are 170 × 130 and 202 × 226, respectively with 51 sigma layers in vertical (12 layers below 1.5 km AGL) between the surface and the model top at 50 hPa. The CMAQ model is applied to the same domains but with three grid cells removed at each lateral boundary of the WRF domains. 15 vertical layers are selected from the 51 WRF layers, including about 8 layers in the boundary layer and 7 layers in the free troposphere.

Anthropogenic emissions are supplied offline from the MIX inventory (Li et al., 2017). Terrestrial biogenic emissions are calculated offline using MEGAN v2.04 (Guenther et al., 2006). Sea salt emissions are incorporated into the AERO4 aerosol module, and calculated online in CMAQ. Wind-blown dust is derived online from the WRF-Dust model. Open biomass-burning emissions are not considered here. It should be noted that the anthropogenic emissions are not fixed in this system, but are automatically adjusted every week according to the system performances in the past week and a series of simple and empirical relationships between emissions and concentrations.

The system provides every day a forecast for the next 192 hours. The NCEP Global Forecast System (GFS)'s products at 00 UTC are used for the initial and boundary conditions of the WRF model with a resolution of 0.5-degree and with a 3-hour interval. For the CMAQ model, the boundary conditions are created using idea profiles, and the chemical initial





fields are initialized from the previous forecasting. In addition, hourly averaged observed concentrations of $SO_2$, $NO_2$, CO, $O_3$, PM2.5 and PM10 from 1415 national control air quality-monitoring sites are assimilated into the initial fields using an optimal interpolation method [Lorenc, 1981]. More information on the code can be obtained from Fei Jiang (jiangf@nju.edu.cn) at Nanjing University. Informatio on WRF-CMAQ is also available on the web site http://carbon.nju.edu.cn/cn/ and https://www.epa.gov/cmaq/cmaq-models-0.

### *2.9. WARMS-CMAQ*

The Community Multiscale Air Quality (CMAQ) model is a 3-D Eulerian chemical transport model that explicitly simulates emissions, gas-phase, aqueous, and mixed-phase chemistry, advection and dispersion, aerosol thermodynamics and physics, and wet and dry deposition. A detailed description and an evaluation of the CMAQ model are available in the papers by Byun and Schere (2006), Foley et al. (2010), and Appel et al. (2017). Here the CMAQ version 5.0.2 is adopted and includes the 2005 Carbon Bond (CB05) chemical mechanism (Yarwood et al., 2005) to represent the gas-phase chemistry. The fifth-generation modal CMAQ aerosol model (aero5) is adopted to formulate the aerosol chemistry and dynamics (Carlton et al., 2010).

In this version, CMAQ is used in an off-line mode. It is forced by pre-calculated hourly meteorological fields for the dynamics and by several emissions fluxes for the chemistry. Meteorology fields that drive chemical transport are produced by the Shanghai Meteorological Service (SMS) WRF ADAS Real-time Modeling System (WARMS). The SMS-WARMS has been extensively evaluated and is providing weather predictions in Eastern China. The modelling domain consists of 760 by 600 horizontal grids at 9-km resolution, with 51 layers in the vertical. As a subdomain of the SMS-WARMS run, the CMAQ domain consists of 430 by 370 horizontal grid cells at 9-km resolution. In the vertical, 26 layers are applied.

The anthropogenic emissions are based on monthly HTAP v2 dataset (http://edgar.jrc.ec.europa.eu/htap_v2/) (Janssens-Maenhout et al., 2015) for year 2010. As suggested by operational forecasting results, the HTAP NOx, $SO_2$ emissions are adjusted to account for rapid economic growth in the region. Biogenic emissions are estimated by the MEGAN model version 2.10 (Guenther et al., 2012). Currently, dust and biomass burning emissions are not included.

For the SMS-WARMS model forecasts, the NCEP GFS output at 0.5 degree is used as a background for ADAS data assimilation scheme, which ingests many local observations (e.g. radar and buoys), and to provide lateral boundary conditions. The chemical boundary conditions are currently based on the default vertical profiles of gaseous species and aerosols in CMAQ that represent clean air conditions. For more details, please contact Ying Xie (yxie33@outlook.com) at the Shanghai Meteorological Service. The CMAQ code is available on the US-EPA modeling site (https://github.com/USEPA/CMAQ/).



**Table 2a. Description of the Different Models**

| Model and Institution | Model Documentation | Type of Model | Spatial Domain | Vertical and Horizontal Resolution | Meteo Data | Initial and Boundary Conditions |
|---|---|---|---|---|---|---|
| IFS ECMWF | CAMS | Global On-line | Global | 60 vertical levels<br><br>T511 (40 km) | ECMWF-IFS | IC: previous forecast corrected by data assimilation (analysis) |
| CHIMERE KNMI | Version 2013b | Regional Off-line | 18-50$^0$N 102-132$^0$E | 8 levels (surface to 500 hPa)<br><br>0.25 degree | ECMWF operational data | IC: previous forecast BC: LMDz-INCA (gas and particles), GOCART (mineral dust) |
| WRF-Chem-MPIM | Version 3.6 | Regional On-line | Domain 1: 8S-51N 59-152E<br><br>Domain 2: 18-45N 95-125E | 51 levels (surf. to 10 hPa)<br><br>Domain 1: 60 km x 60 km<br><br>Domain 2: 20 km x 20 km | NCEP-FNL 6 hours 1$^0$ x 1$^0$ | IC: previous forecast BC: IFS |
| SILAM FMI | Version 5.5 | Regional Off-line | 7-54N 67-147E | 14 hybrid sigma-pressure levels up to ~ 400hPa 0.125$^0$ x 0.125$^0$ | ECMWF-IFS | IC: previous forecast BC: Silam global forecast |
| EMEP MET Norway | Svn3064 | Regional Off-line | 15-55N 90-135E | 20 sigma levels (surf. to 50 hPa) | ECMWF-IFS | IC: previous forecast BC: ECMWF IFS (3-hourly) |
| LOTOS-EUROS | Version 1.10 | Regional Off-line | Domain 1: 15-50 N 71-139 E | 5 layers (surf. to 5 km) | ECMWF-IFS | IC: previous forecast BC: CAMS |





| | | | | Domain 1: 0.5⁰ x 0.25⁰ | | C-IFS (3-hourly) |
|---|---|---|---|---|---|---|
| | | | Domain 2: 20-45N 105-130ᴱ | Domain 2: 0.25⁰ x 0.125⁰ | | |
| **WRF-Chem SMS** | Version 3.2 | Regional On-line | 20-44N 110-126E | 28 vertical layers (surf. to 50 hPa)  6 km | NCEP GFS 6 hours 0.5⁰ x 0.5⁰ | IC: Previous run BC: MOZART monthly averages for 2009 |
| **WRF-CMAQ NJU** | WRFv3.5 CMAQv 4.7.1 | Regional Off-line | Domain 1: 18-52N, 78-136E Domain 2: 21-44N, 102-125E | Domain 1: 36 km x 36 km Domain 2: 12 km x 12 km WRF: 51 sigma levels CMAQ: 15 sigma levels | NCEP GFS 3 hours 0.5⁰ x 0.5⁰ | IC: Previous run BC: CMAQ default vertical profile |
| **WARMS-CMAQ SMS** | Version 5.0.2 | Regional Off-line | 14-53 N 100-144 E | 26 sigma levels (from surf. to 50 hPa)  9 km | NCEP GFS 6 hours 0.5⁰ x 0.5⁰ | IC: Previous run BC: CMAQ default vertical profile |

**Table 2b.  Continued**

| Model and Institution | PBL | Land-Use | Deposition | Chemistry | Data Assimilation |
|---|---|---|---|---|---|
| **IFS ECMWF** | IFS PBL scheme | IFS-Land use | Dry: Resistance Wet: in-cloud and below cloud scavenging and evaporation | Gas: CB05 Aerosol: LMDz/MACC | yes $(O_3, CO, NO_2, SO_2, HCHO)$ |
| **CHIMERE KNMI** | bulk Richardson number (Menut et al., 2013) | GlobCover LandCover verion 2.3, 2009 | Dry: Resistance Wet: in-cloud and below cloud scavenging | gas: MELCHIOR2 aerosol: Schemes for nucleation, absorption(ISO RROPIA), and | no |





| | | | | | |
|---|---|---|---|---|---|
| | | | | | coagulation |
| **WRF-Chem-MPIM** | YSU | MODIS | Dry: Resistance Wet: in-cloud scavenging | gas: MOZART4 aerosol: GOCART | no |
| **SILAM FMI** | Bulk-Rishardson number, modified to use 2t and U*. | Maps of roughness, LAI from C-IFS | Dry: Resistance for gases, Kouznetsov&Sofiev (2012) for particles Wet: Rainout and washout with air-water equilibria | gas: CBM-IV aerosol: DMAT/VBS | not used |
| **EMEP MET Norway** | Slightly modified bulk Richardson number, PBL height always between 100-3000 m | GLC2000 | Dry: Resistance Wet: in-cloud and below cloud scavenging | MARS module for aerosols<br><br>Gas: EmChem09 | no |
| **LOTOS-EUROS** | Version 1.10 | Regional Off-line | Domain 1: 15-50 N 71-139 E<br><br>Domain 2: 20-45N 105-130$^E$ | 5 layers (surf. to 5 km)<br><br>Domain 1: $0.5^0$ x $0.25^0$<br><br>Domain 2: $0.25^0$ x $0.125^0$ | ECMWF-IFS |
| **WRF-Chem SMS** | YSU | MODIS | Dry: Resistance Wet: in-cloud scavenging | gas:RADM2 aerosol: ISORROPIA/SORGAM | no |
| **WRF-CMAQ NJU** | YSU | USGS modified with MODIS urban cover data | Dry: Resistance Wet: in-cloud and below cloud scavenging | Gas: CB05 Aerosol: aero4 | Yes ($SO_2$, $NO_2$, CO, $O_3$, PM2.5, PM10) |
| **WARMS-CMAQ SMS** | YSU | MODIS | Dry: Resistance Wet: in-cloud and below cloud scavenging | gas: CB05 aerosol: CMAQ aero5 | no |


## 3. Adopted Emissions

The choice of the adopted surface emissions for primary chemical species has a significant
influence on the atmospheric concentrations calculated for these species and for related
secondary pollutants. In this inter-comparison exercise, the different groups involved have





adopted their preferred anthropogenic emissions based on published inventories such as
MEIC (Li et al., 2014; Liu et al., 2015), MACCity (Granier et al., 2011), EDGAR (Muntean et al.,
2014; Crippa et al., 2016) and HTAP (Janssens-Maenhout et al., 2015). An inventory
developed specifically for the PANDA project called PanHam has been obtained by
combining information from the MEIC and HTAP inventories.  Each model uses its own
formulation for dust mobilization or seal salt emissions. In most cases, the biogenic
emissions are derived online or offline from the MEGAN model (Guenther et al., 2006, 2012).
Table 3 provides more details about the specified emissions and Figure 1 shows the mean
distribution of the anthropogenic emissions for CO, NO and $SO_2$ adopted by different
models during the period 1-14 March 2017. In the case of carbon monoxide, the adopted
emissions are relatively similar in all models with mean emissions ranging from 4.0 to 4.6
mg m$^{-2}$ h$^{-1}$. In the case of nitric oxide, however, there are substantial differences with mean
emissions ranging from 0.31 mg m$^{-2}$ h$^{-1}$ (WRF-Chem-MPIM) to 0.99 mg m$^{-2}$ h$^{-1}$ (EMEP), but
with values around 0.30 – 0.45 mg m$^{-2}$ h$^{-1}$ used by most models. For sulphur dioxide,
produced primarily from coal combustion, the adopted values range from 0.31 mg m$^{-2}$ h$^{-1}$
(WRF-Chem-SMS) to 0.73 mg m$^{-2}$ h$^{-1}$ (IFS), but with values around 0.67 mg m$^{-2}$ h$^{-1}$ adopted in
most models. The low values adopted for WRF-Chem-SMS reflect the likely impact of the
recent measures taken in China to limit the emissions from coal burning facilities.
Emission inventories that are currently available to the modelling community usually
account for anthropogenic emissions for years 2010 to 2012, and hence do not account for
the substantial reduction in the emissions that took place since around 2014 as a result of
actions taken by the Chinese authorities. The lower emission values adopted by several
models may therefore be more realistic for providing chemical weather forecasts in 2017.

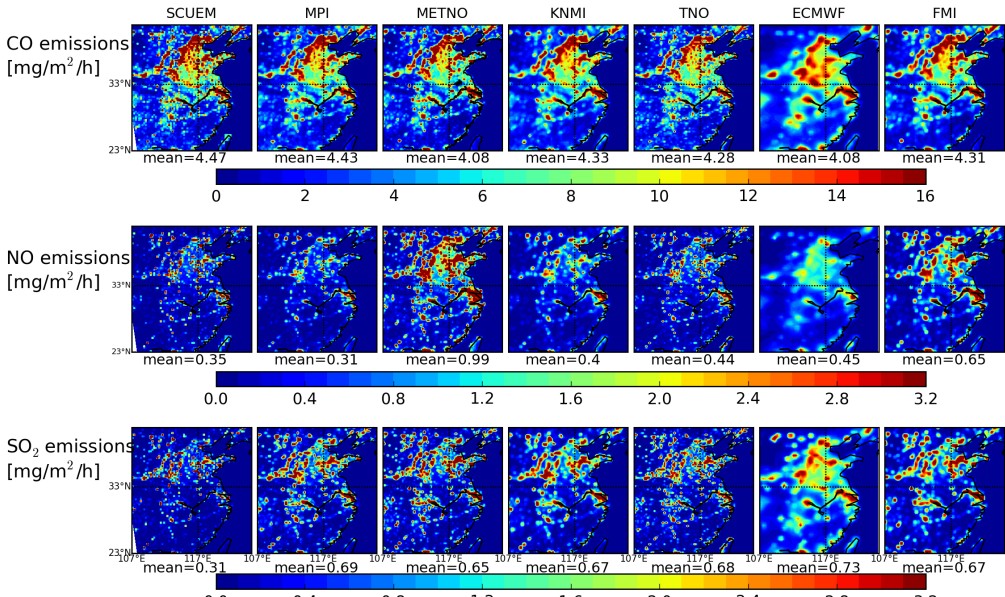




**Figure 1.** Surface emissions of CO, NO and $SO_2$ [mg m$^{-2}$ h$^{-1}$] adopted by the different models (average
for the period 1-14 March 2017). Note that the SCUEM emissions are those used in the WRF-Chem-
SMS model.
**Table 3. Adopted Emissions**

| Model and Institution | Anthro. dataset | Dust | Seasalt | Biogenic | Biomass burning | Special Treatment/ Modification |
|---|---|---|---|---|---|---|
| IFS ECMWF | MACCity | Ginoux et al (2001) | Monahan et al. (1986) | Monthly climatology of MEGAN v2 run | GFAS | Diurnal cycle for isoprene |
| CHIMERE KNMI | MEIC 2010 | none | none | MEGAN | none | none |
| WRF-Chem-MPIM | HTAPv2 | GOCART | MOSAIC | MEGAN | none | Diurnal profiles by sector; Anthro NOx emission -50%; |
| SILAM FMI | MACCity with excluded Shippig, STEAM2015 Shipping, PanHam for Coarse PM | SILAM Scheme after Zender (2003) | SILAM Scheme Sofiev et al (2012) | MEGAN-MACC | GFAS (gases), IS4FIRES (PM) | Diurnal profiles by sector |
| EMEP MET Norway | PanHam (HTAP + MEIC2012) | none | Tsyro et al. (2011) | EMEP scheme | GFAS | none[1] |
| LOTOS-EUROS | EDGAR + MEIC2010 | online | online | MEGAN | GFAS | Anthro NOx emission -35%; Anthro SO2 emission -50% |
| WRF-Chem SMS | MEIC 2010 | With dust BC from WRF-Dust | none | MEGAN v2 | none | Diurnal profiles by sector; |

---

[1] Non during the inter-comparison exercise. Since summer 2017, however, the NOx emissions have been reduced by 35% in this particular model. The present version of the model also calculates windblown dust emissions from soil erosion.



| | | | | | | Anthro NOx emission -40%; Anthro SO$_2$ emission -60% |
|---|---|---|---|---|---|---|
| **WRF-CMAQ NJU** | MIX | WRF-Dust | CMAQ scheme | MEGAN v2.04 | none | Adjusted by performance of last week |
| **WARMS-CMAQ SMS** | HTAPv2 | none | CMAQ scheme | MEGAN v2.10 | none | Diurnal profiles by sector; Anthro NOx emission -50%; Anthro SO$_2$ emission -70% |


## 4. Operational Forecasts provided by the MarcoPolo-Panda System.

As stated above, the MarcoPolo-Panda system is used operationally to provide daily
forecast of air quality in eastern China. In its present configuration (Figure 2), the system is
based on 9 models, which are executed independently on the computing system available in
each respective partner institution. The outputs of the models are locally processed and the
surface concentrations of the key chemical species are forwarded to a central database
operated by the Royal Netherlands Meteorological Institute (KNMI). Ensemble mean and
median concentrations are derived and, in addition to the forecasts from individual models,
are posted on a dedicated website (www.marcopolo-panda.eu) and Chinese mirror site
(http://116.62.195.108/). For the 37 Chinese cities with a population above 3 million in 2010,
the predicted concentration values of ozone, NO$_2$, PM2.5 and PM10 are compared each
hour to local measurements reported by the Chinese monitoring network (www.pm25.int).
Observations for each city represent the mean of several measurements performed within
one city (usually 5-12 stations). The data are averaged to city-centre coordinates.
We start by presenting a few examples of randomly selected forecasts as provided by the
MarcoPolo-Panda system to illustrate the diversity among the models and the differences
obtained under different situations. The performance of each individual model varies from
day to day because it strongly depends on the individual weather forecast (meteorological
situation, cloudiness, precipitation, etc.) that is adopted to simulate transport,
photochemistry and deposition. Therefore this first description of model forecasts does not
provide reliable information on the accuracy of the forecasts provided by the different
models included in the ensemble.



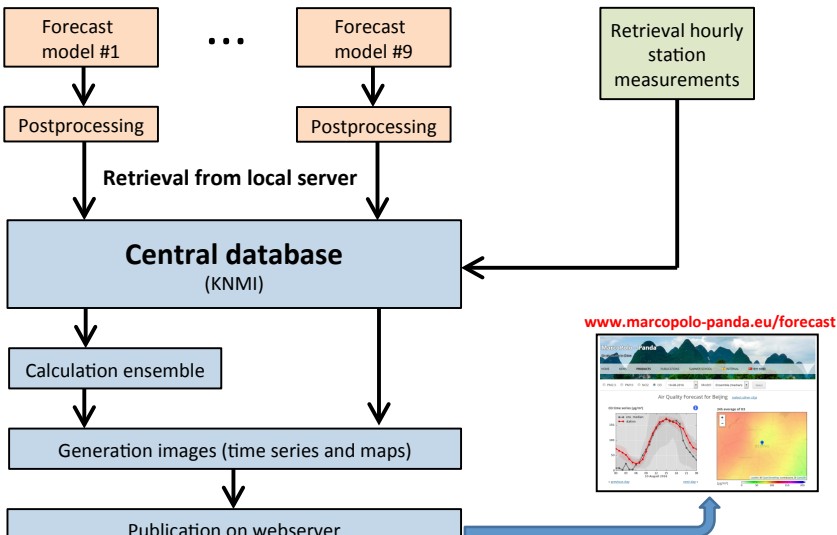

**Figure 2.** Structure of the operational multi-model forecast system with the 9 model components.
Postprocessed forecasts for the next 3 days provided by each model are sent to a central database
maintained by the Royal Netherlands Meteorological Institute (KNMI). Ensemble medians and
means are calculated and information (predicted daily variations of surface concentrations for 37
major Chinese cities, and maps of predicted diurnal mean surface concentrations) and are posted on
the http://www.marcopolo-panda.eu/forecast website. Users in China are redirected to the mirror
website maintained by SMS (http://116.62.195.108/). The forecasts are compared with the median
and mean observations provided by monitoring stations at different locations of the 37 cities.
The first example presents a relatively successful forecast made for the coastal city of
Xiamen in southeast China on 13 October 2017. The panels in Figure 3 show the excellent
agreement in the case of $NO_2$, ozone and PM2.5, suggesting that the median values derived
from the individual models capture well the features associated with the meteorological
situation, atmospheric transport and with the emissions in the region on that particular day.
The situation corresponds to very clean conditions with PM2.5 and $NO_2$ concentrations of
the order of 10 - 15 µg m$^{-3}$. The predicted ozone concentration ranges from 70 - 90 µg m$^{-3}$
(35 to 45 ppbv). Interestingly, however, the predicted PM10 concentrations are
underestimated during most of the day. The model predicts concentrations close to 20-25
µg m$^{-3}$, while the measurements indicate that the concentration reached values as high as
30-40 µg m$^{-3}$. The presence on October 13 of a strong wind flow in the strait between
Mainland China and Taiwan and associated with the Khanun tropical depression present on
this particular day west of the Philippines was likely a source of elevated sea salt emissions
and dust mobilization that may not have been properly captured by the models. Under such
strong meteorological disturbance, the forecast could be strongly resolution dependent.






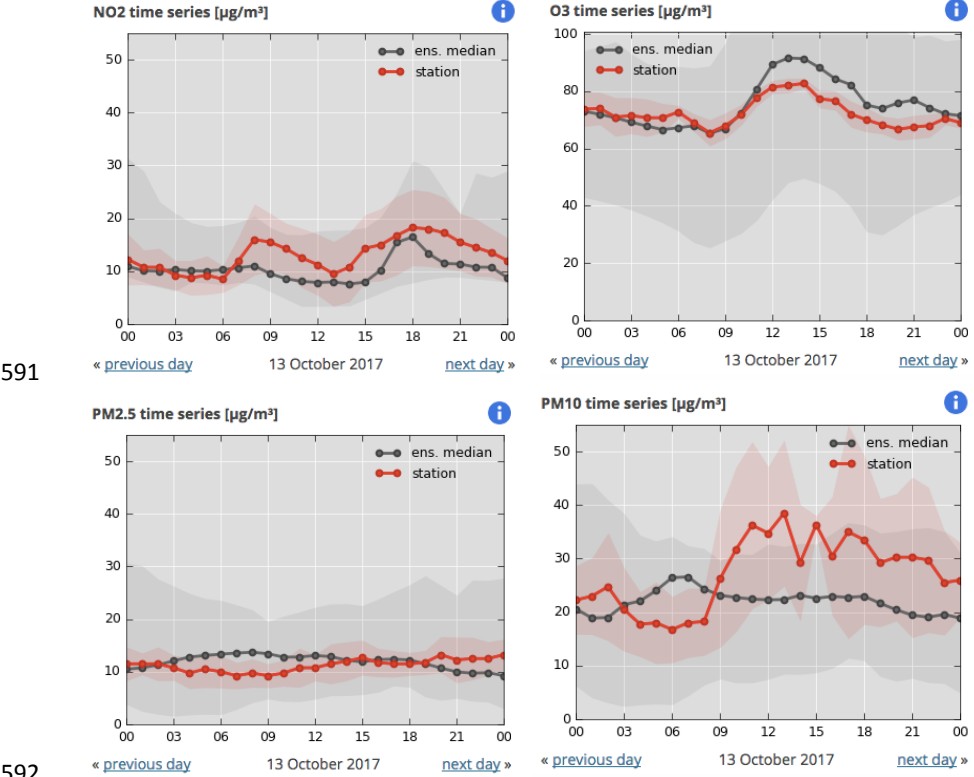

**Figure 3.** Median concentrations of $NO_2$ (upper, left), ozone (upper, right), PM2.5 (lower, left) and
PM10 (lower, right) predicted for the city of Xiamen on 13 October, 2017 (black curve) and
compared with the measured values (red curves). The dispersion of the forecasts by the individual
models belong to the ensemble is shown by the grey range and the dispersion of the measured
values at different stations in the city are depicted by the pink band.

The second example of predictions (Figure 4) refers to the forecast of $PM_{2.5}$ in Shanghai on a
relatively polluted day (3 November, 2017). All models predict the presence of relatively
high concentrations over land (diurnal mean values of typically 100 -150 µg m$^{-3}$) with a
steep negative gradient towards the Chinese sea, where the concentrations are of the order
of only 25-40 µg m$^{-3}$. Observations made at different stations in this urban area show the
occurrence of two successive concentration peaks, one around 9:00-10:00 with
concentrations reaching about 180 µg m$^{-3}$ and the second one at 15:00-16:00 with
concentrations as high as 150 µg m$^{-3}$. The ensemble mean forecast system predicts the
occurrence of a single peak at about 7:00 am with a $PM_{2.5}$ concentration of about 220 µg m$^{-}$
$^{3}$. The forecast shows a gradual decrease in the concentration during the afternoon that is in
good agreement with the observation. The occurrence of the second peak in the afternoon,
however, is missed by the ensemble prediction, even though a peak appears in some of the
individual model calculations (WRF-Chem SMS, EMEP and WRF-CMAQ), but often a few
hours before it was actually detected by the monitoring stations. An inspection of the
forecasts by the different models highlights the diversity in the model results. IFS, CHIMERE,
WRF-Chem-SMS, and EMEP overestimate the PM2.5 concentrations before mid-day, while



they provide values in good agreement with the observations in the afternoon and evening.
WRF-Chem-MPIM underestimates the concentrations during the entire day. LOTOS-EUROS
as well as WRF-CMAQ provide values that are in fair agreement with the observations in the
morning, but underestimate the concentrations in the afternoon.

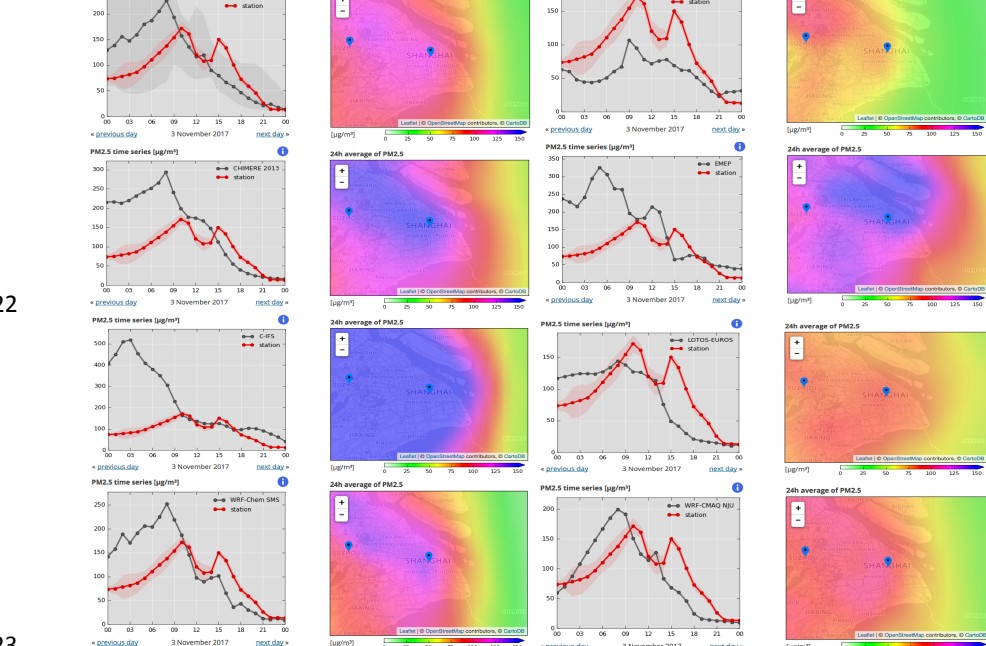


**Figure 4.** Forecast by different models of PM$_{2.5}$ concentration during a polluted day in Shanghai on 3
November 2017. The graph at the top left represents the median concentration, and the individual
forecasts provided by CHIMERE, IFS, WRF-Chem-SMS, WRF-Chem-MPIM, EMEP, LOTOS-EUROS, and
WRF-CMAQ are shown by the other panels.
A third example (Figure 5) refers to the predicted concentration of PM$_{2.5}$ on 25 October
2017 in Beijing. In this particular case, the ensemble forecast system predicts the
occurrence of a rather polluted day with stagnant air and high concentrations of aerosol
particles over Beijing as a band stretching from the southwest to the northeast. The median
concentration predicted for this day is close to 200 µg m$^{-3}$, but is a factor 2 higher than the
observation. Most individual models produce this band of high PM2.5 concentrations with
the exception of the WRF-Chem-MPIM model that shows moderate levels of pollution with
an aerosol cloud localized in the urban area of Beijing. An examination of the results
provided by the individual models shows again large differences. Some models (CHIMERE,
EMEP, LOTOS-EUROS, WRF-Chem-MPIM) calculate a slow and rather steady concentration
increase during the day, while other models (WRF-Chem-SMS, WARMS-CMAQ-SMS, SILAM
and IFS) exhibit some irregular variations during the day. Most models overestimate the
PM$_{2.5}$   concentrations   except   LOTOS-EUROS   and   WRF-Chem-MPIM,   which   predict



concentrations with the same order of magnitude as the observations at the monitoring
stations.





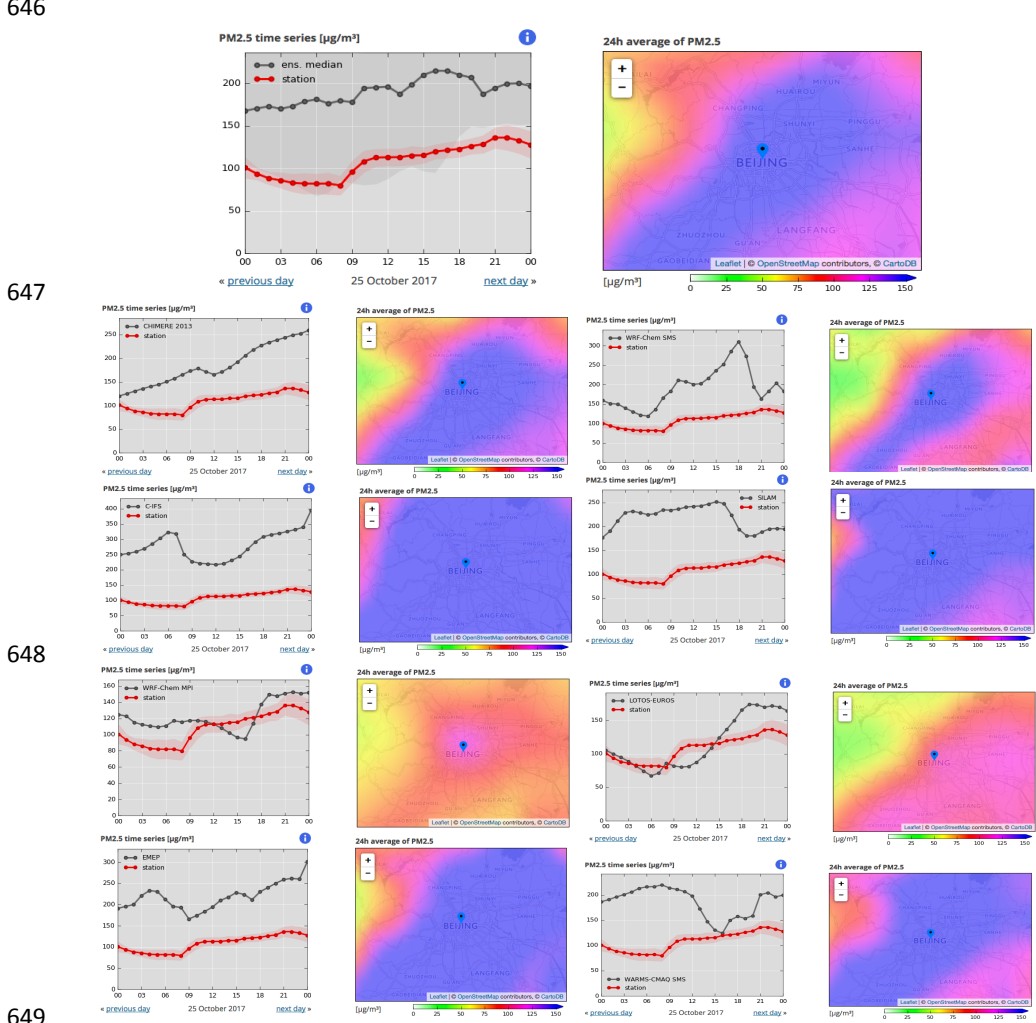



**Figure 5.** Diversity of PM$_{2.5}$ forecasts in Beijing on 25 October 2017 by several models included in
the ensemble of the MarcoPolo-Panda prediction system. The ensemble median is shown by the top
panels, and the individual forecasts provided by CHIMERE, IFS, WRF-Chem-MPIM, EMEP, WRF-
Chem-SMS, SILAM, LOTOS-EUROS, and WARMS-CMAQ-SMS are shown by the other panels.
The last illustrative example refers to the forecast of nitrogen oxides and ozone in the
Shanghai area on 31 October 2017 (Figure 6a, b and c). All models show that the NO$_2$
concentrations are highest in the boundary layer of the urban areas, even though the
calculated values may be different from model to model, and the dispersion of the species
away from the urban centres may also be uneven. In all cases, predicted values above the
ocean are very low, i.e., less than a few µg m$^{-3}$. A band of high NO$_2$ concentrations extends
from Shanghai in the northwest direction.





The median values of $NO_2$ in the city (top panels) are in good agreement with the observed
values, with night-time concentrations on the order of 60-80 µg m$^{-3}$, and substantially lower
values during daytime resulting from the photolysis of the molecule by solar radiation. A
minimum concentration of 25 µg m$^{-3}$ is reached around noon.

The diurnal variation of $NO_2$ is well captured by most models, in particular by CHIMERE
(although the absolute values are too low), IFS, WRF-Chem-SMS, WRF-Chem-MPIM and
WARMS-CMAQ-SMS. The diurnal variation is somewhat underestimated in EMEP, LOTOS-
EUROS and WRF-CMAQ.

The ozone concentration (right panels) also exhibits a strong diurnal variation that, to a
large extent, mirrors the $NO_2$ variation. Measurements show a maximum value of nearly
100 µg m$^{-3}$ reached at 15:00 and low night-time concentrations (typically 10-30 µg m$^{-3}$). The
median concentrations, provided by the ensemble forecast system upper panel on the
right), are characterized by a similar diurnal variation but with lower amplitude. The
concentration reaches its maximum at 14:00, but the value of this maximum is only equal to
60 µg m$^{-3}$. The values predicted for the night are generally somewhat smaller than the
observation, with values of the order of 5-10 µg m$^{-3}$.

In the case of ozone, differences between model forecasts are again substantial. The
maximum concentration values in the early afternoon are 50 µg m$^{-3}$ for CHIMERE, 62 µg m$^{-3}$
for IFS, 85 µg m$^{-3}$ for WRF-Chem-SMS, 65 µg m$^{-3}$ for WRF-Chem-MPIM, 30 µg m$^{-3}$ for EMEP,
42 µg m$^{-3}$ for LOTOS-EUROS, 57 µg m$^{-3}$ for WRF-CMAQ and 100 µg m$^{-3}$ for WARMS-CMAQ-
SMS.

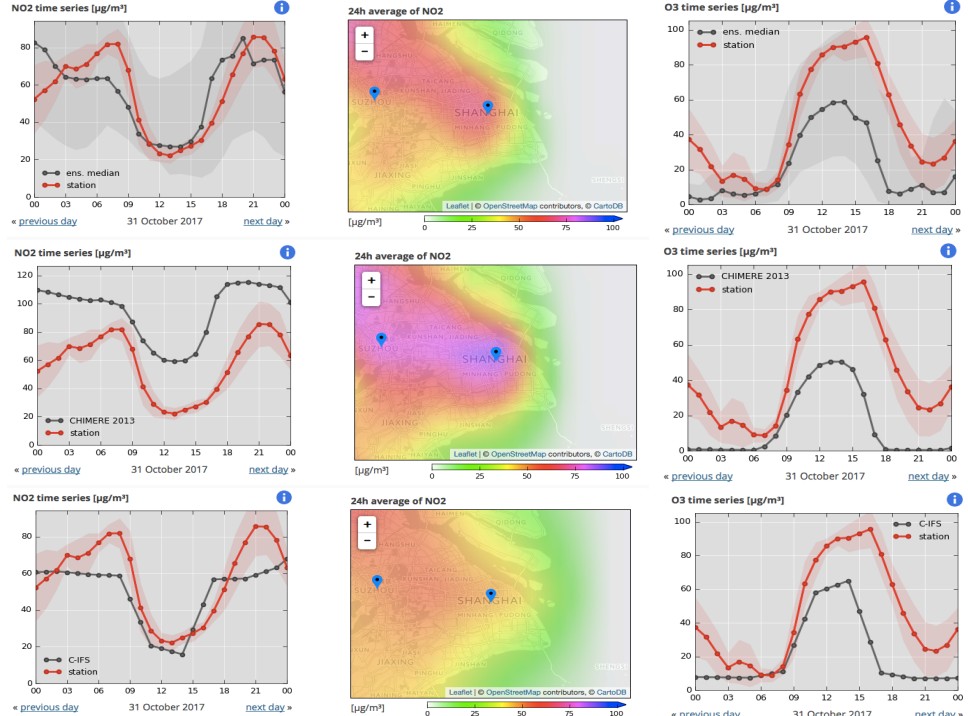

**Figure 6a.** Diversity in the NO$_2$ and ozone forecasts made for Shanghai on 31 October 2017 as highlighted by the predictions from several models included in the ensemble of the MarcoPolo-Panda system. The left and right panels show the diurnal variation of the predicted (black) and observed (red) NO$_2$ and ozone concentrations ($\mu$g m$^{-3}$), respectively. The center panel presents the geographical distribution in the vicinity of Shanghai of the diurnal average predicted for the NO$_2$ concentration. The ensemble median is shown in the top panels, and two individual forecasts as provided by CHIMERE and IFS are shown in the middle and lower panels.





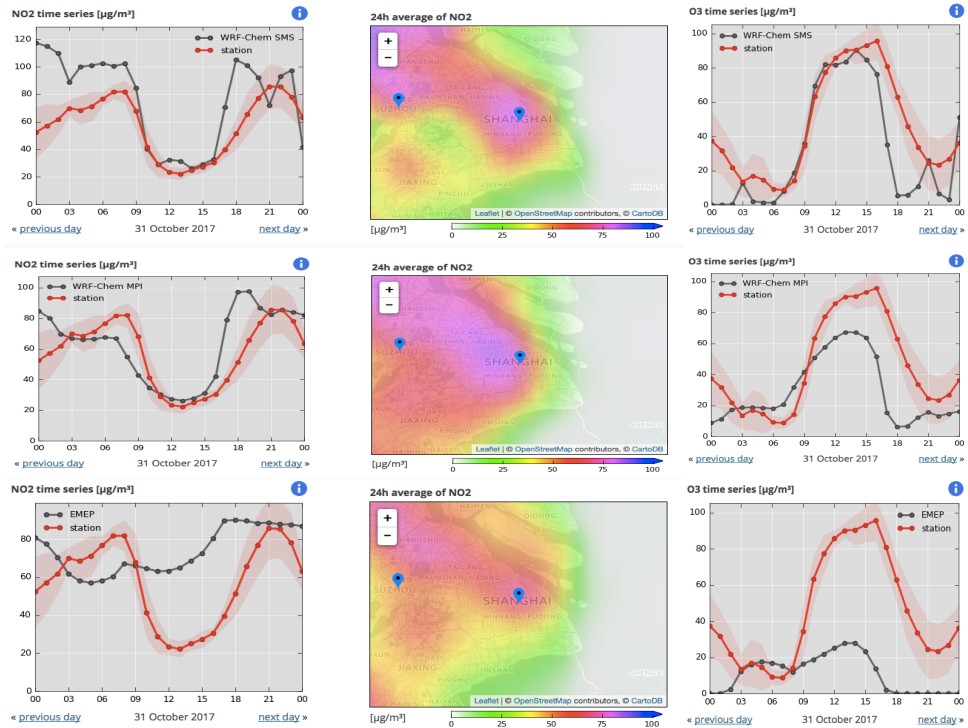

**Figure 6b.** Same as in Figure 6a, but for the individual forecasts from WRF-Chem-SMS, WRF-Chem-
MPIM and EMEP.





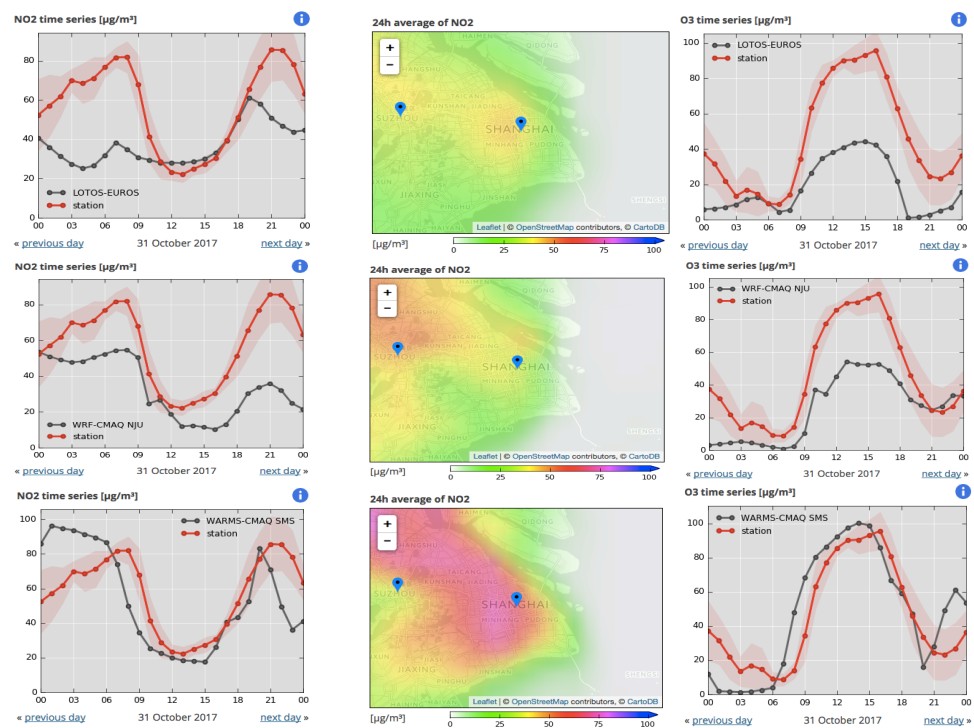

**Figure 6c.** Same as Figure 6a but for the individual forecasts from LOTOS-EUROS, WRF-CMAQ and
WARMS-CMAQ.






**5. Inter-comparison of Individual Models**

We now present an inter-comparison of most of the models included in the operational
MarcoPolo-Panda System. The participants to this inter-comparison examined in detail the
daily forecasts performed for the month of March 2017 with particular emphasis on the
results obtained during the first two weeks of the month.

In the following Sections, we present selected chemical fields derived by the different
models that participated in the comparison exercise, and highlight similarities and
differences with the purpose of identifying the causes of the discrepancies between models
and between models and observations. We first examine monthly mean surface
concentrations obtained from a subset of the models involved in the inter-comparison. We
then compare the time evolution associated with the model forecasts with observations
made at specific surface measurement sites and present some correlations between
calculated and measured concentrations at these sites.

*5.1.  Comparison of average fields*


We first compare the March 2017 monthly mean concentrations of different chemical
species calculated by 7 models (IFS, LOTOS-EUROS, EMEP, SILAM, WRF-Chem-MPIM, WRF-
Chem-SMS and CHIMERE) with surface measurements reported at different sites in the
eastern part of China (www.pm25.int). Figure 7a shows the calculated and observed surface
concentrations of carbon monoxide (CO).  We first note the substantial differences that
exist between the individual model forecasts, probably reflecting differences in the adopted
emissions or in the atmospheric production resulting from the oxidation of volatile organic
compounds in the planetary boundary layer. Observations indicate that CO concentrations
are generally higher than 900 ppbv, except near the south-eastern coast and in the south-
western part of the country, where the values are as low as 500 to 700 ppbv. The models
show considerably lower values, ranging from about 300-500 ppbv. The regions with the
highest mean concentrations are located in the North China Plain (NCP), where values
higher than 1200 ppbv are recorded. Relatively high values (close to 1000 ppbv) are also
found in some urban areas (e.g., Hong Kong) near the south coast of the country.

The models provide a rather different picture: most of them substantially underestimate the
CO concentrations, in particular WRF-Chem-SMS, WRF-Chem-MPIM, EMEP and LOTOS
EUROS. Higher concentrations are derived by SILAM and IFS. These models, however,
produce peak concentrations in the region of Sichuan Basin in contrast with the
observations. Only IFS reproduces the high concentrations observed in northern China.
Clearly, the performance of the models regarding the calculation of CO concentrations is not
satisfactory. The discrepancies may be attributed to an underestimation of CO emissions,
errors in the lateral boundary conditions or indirectly to an underestimation of the
emissions for primary hydrocarbons.



We first compare the March 2017 monthly mean concentrations of different chemical
species calculated by 7 models (IFS, LOTOS-EUROS, EMEP, SILAM, WRF-Chem-MPIM, WRF-
Chem-SMS and CHIMERE) with surface measurements reported at different sites in the
eastern part of China. Figure 7a shows the calculated and observed surface concentrations
of carbon monoxide (CO). We first note the substantial differences that exist between the
individual model forecasts, probably reflecting differences in the atmospheric production of
CO resulting from the oxidation of volatile organic compounds or from the chemical
destruction of CO occurring in the planetary boundary layer. Observations indicate that CO
concentrations are generally higher than 900 ppbv, except near the south-eastern coast and
in the south-western part of the country, where the values are as low as 500 to 700 ppbv.
The regions with the highest mean observed concentrations are located in the North China
Plain (NCP), where values higher than 1200 ppbv are recorded. Relatively high values (close
to 1000 ppbv) are also found in some urban areas (e.g., Hong Kong) near the south coast of
the country.
The models provide a rather different picture: most of them substantially underestimate the
CO concentrations, in particular WRF-Chem-SMS, WRF-Chem-MPIM, EMEP and LOTOS
EUROS with calculated values ranging from about 300-500 ppbv. Higher concentrations are
derived by SILAM and IFS. These models, however, produce peak concentrations in the
region of Sichuan Basin in contrast with the observations. Only IFS reproduces the high
concentrations observed in northern China. Clearly, the performance of the models
regarding the calculation of CO concentrations is not satisfactory. The discrepancies may be
attributed to an underestimation of CO emissions, errors in injection height or
overestimation of mixing/boundary layer height, errors in the lateral boundary conditions or
indirectly an underestimation of the emissions for the primary hydrocarbons.

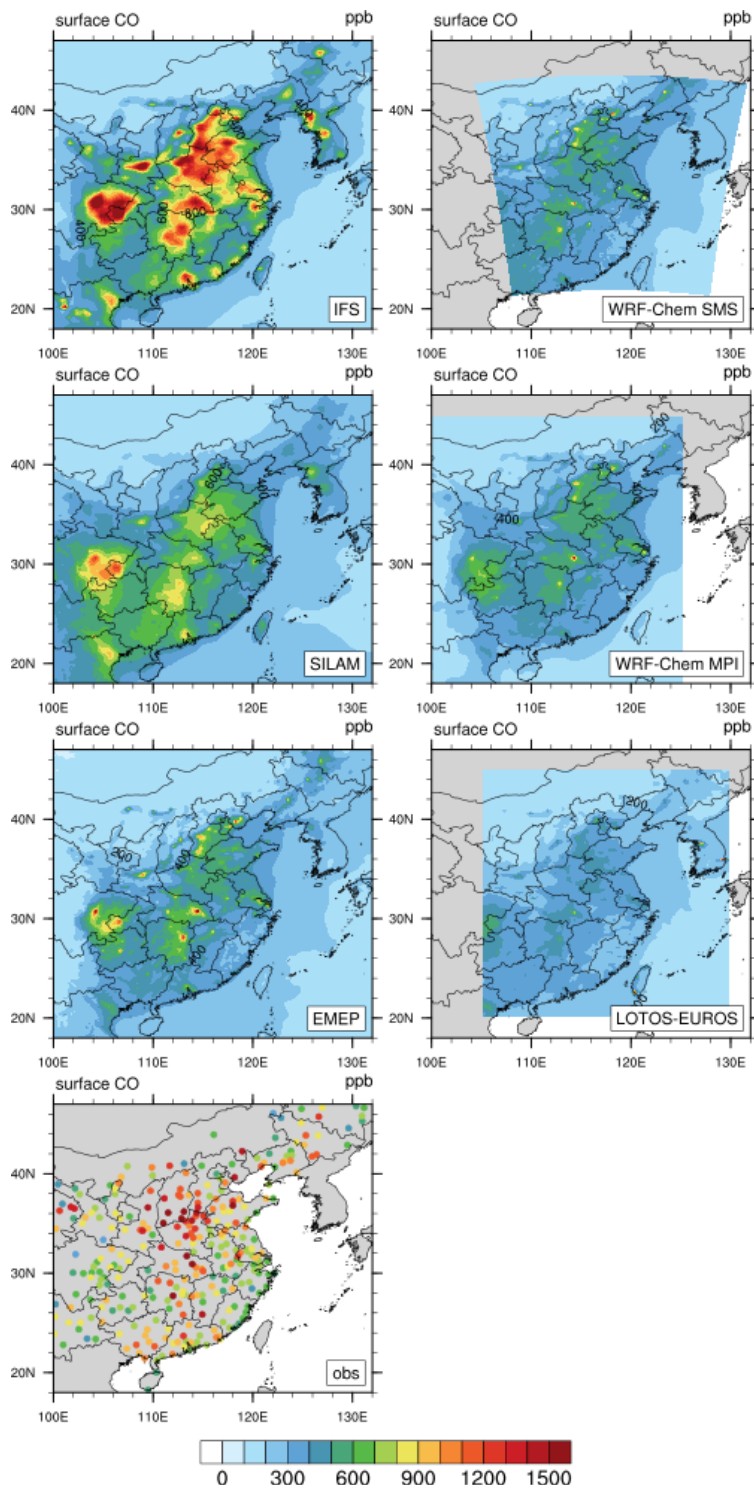





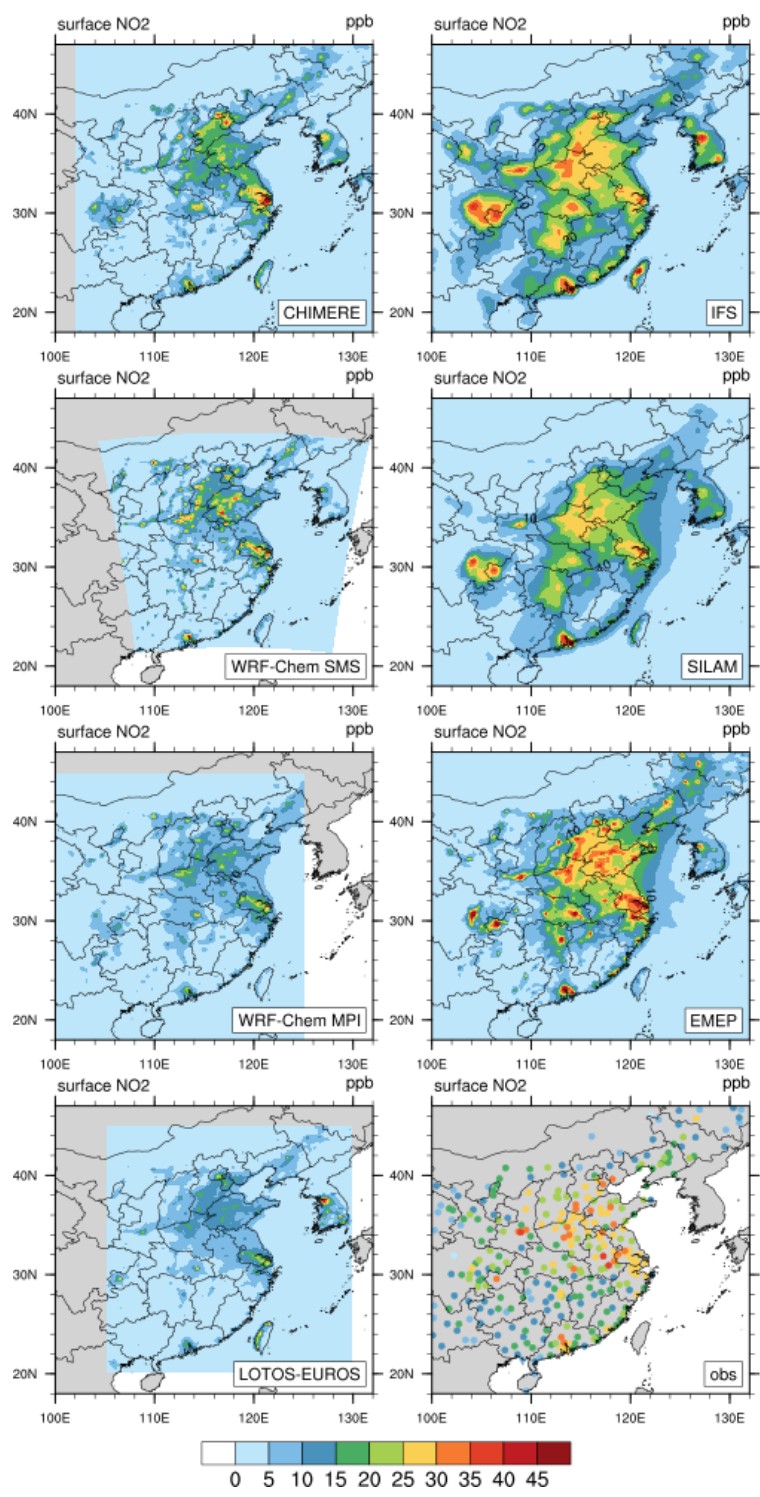





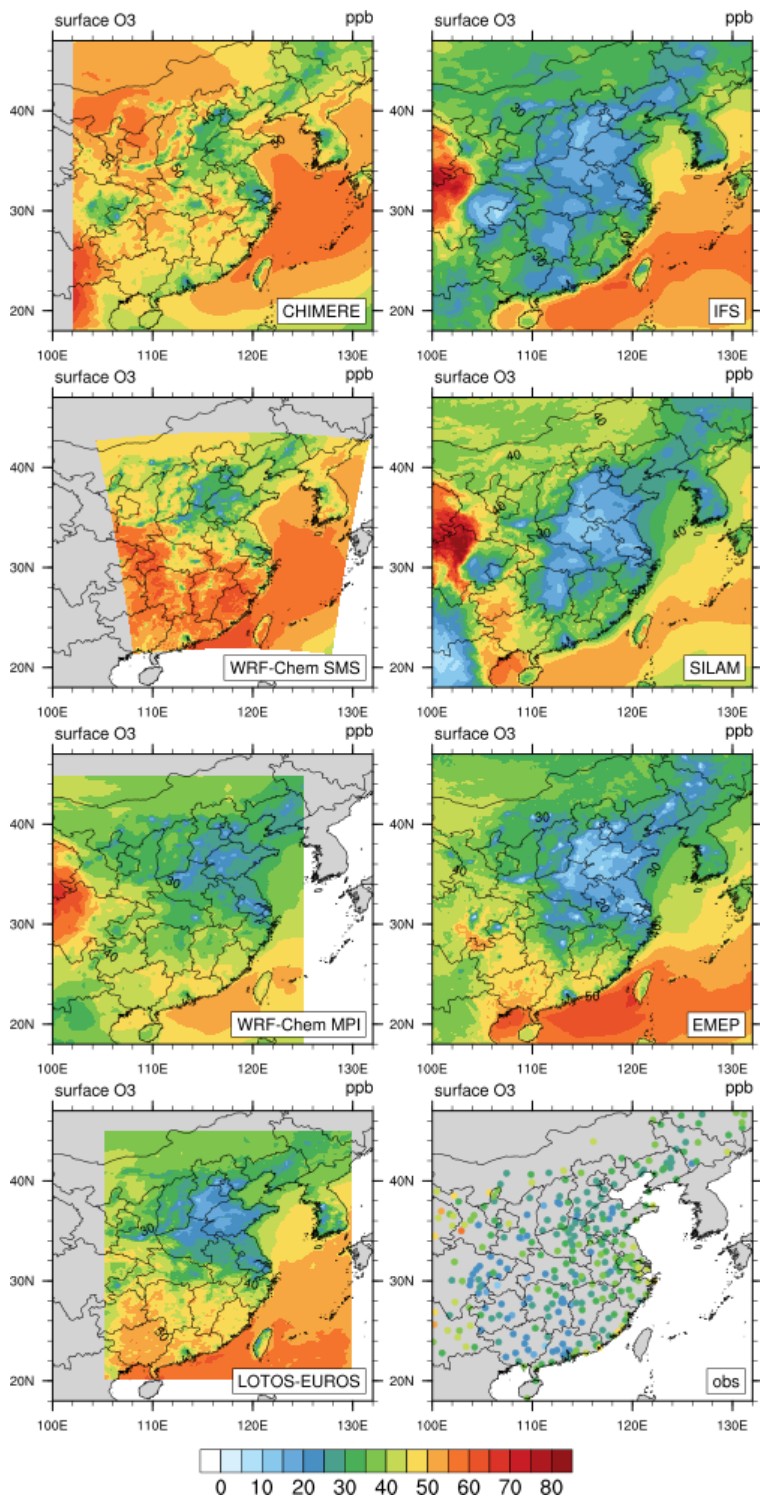





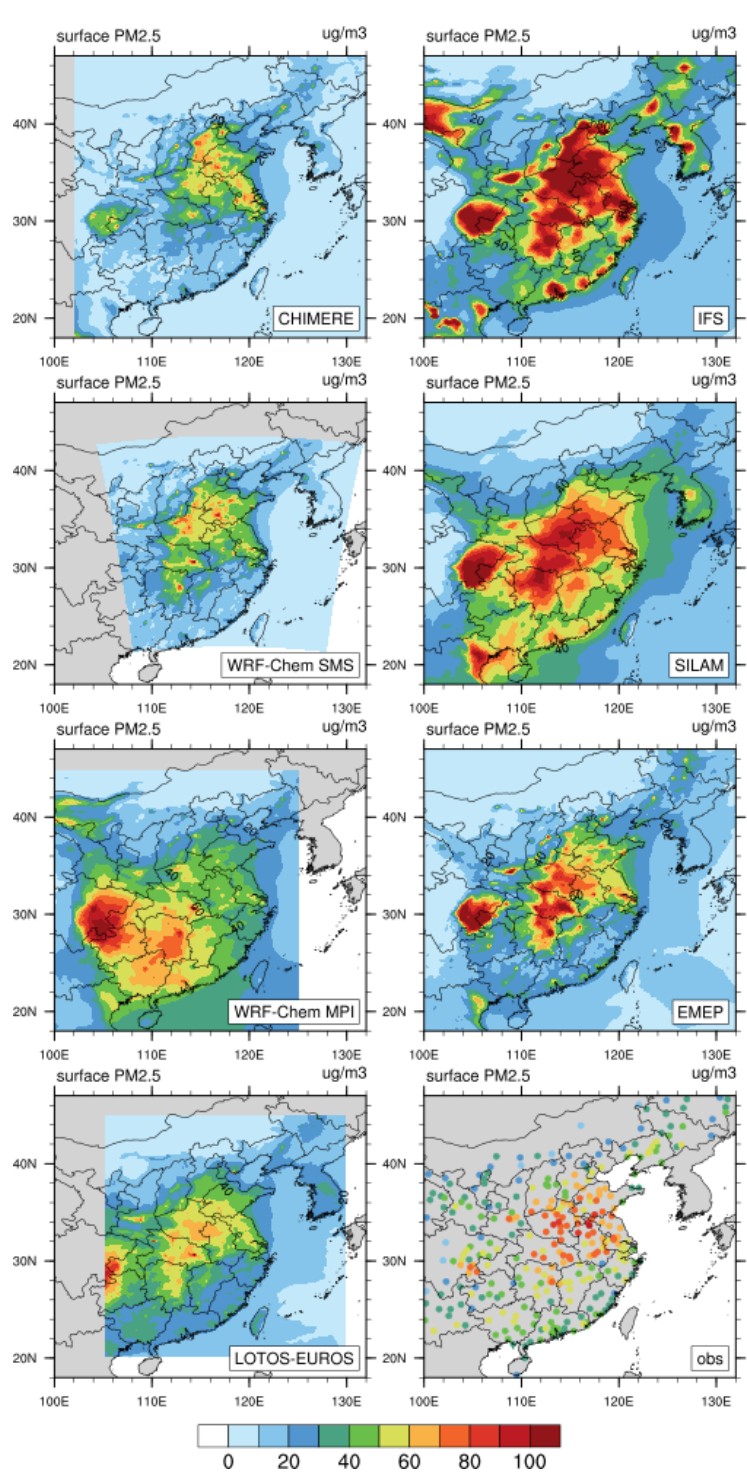




**Figure 7.** Monthly mean surface concentrations of CO, $NO_2$, ozone (ppbv), and PM2.5 ($\mu g\ m^{-3}$)
provided for the month of March 2017 by different models: CHIMERE (no CO), IFS, WRF-Chem-SMS,
SILAM, WRF-Chem-MPIM, EMEP and LOTOS-EUROS. The monthly mean concentration values
derived from observations at different monitoring stations are represented by dots in one of the
lowest panels. The adopted colour scales are the same as the colour scales adopted to represent the
model results.
In the case of $NO_2$ (Figure 7b), the observations show that the surface concentrations are
highest in the north-eastern portion of China with a few urban hotspots. These patterns are
well reproduced by the EMEP, SILAM and IFS models. The other models also produce high
concentrations in urban areas, but with values that are lower than those provided by the
monitoring stations.
The mean surface ozone concentrations derived from measurements are lowest (about 20
ppbv) in the central part of China and highest (30-40 ppbv) near the east coast (Shanghai
region), the south coast and the western part of China. Since nitrogen oxides tend to titrate
ozone, the models that predict high $NO_2$ concentrations derive the lowest ozone values
(EMEP, SILAM, IFS). The high $NO_2$ concentrations predicted by EMEP are probably related to
the large emissions used as shown in Fig 1. CHIMERE, WRF-Chem-SMS and to a lesser extent
WRF-Chem-MPIM overestimate the mean ozone concentration during March. All models,
however, produce a minimum in the ozone concentrations in north-eastern China, a pattern
that is not visible in the observational data (Figure 7c).
Finally, in the case of PM2.5 (Figure 7d), the measurements suggest the presence of high
concentrations (higher than 80 $\mu g\ m^{-3}$) in the region between Beijing and Shanghai. High
abundances of PM2.5 are derived in this region by IFS, SILAM and to a lesser extent by
LOTOS-EUROS, EMEP, CHIMERE and WRF-Chem-SMS. Interestingly, most models produce
another marked hotspot in the region of Sichuan Basin, while the observations suggest a
less pronounced maximum with a more limited geographical extent.
### 5.2. Time Evolution of Median Forecasts
We now focus on the time period during which the most intensive comparison between
models has been performed. We first examine the time evolution of surface ozone, $NO_2$ and
$PM_{2.5}$ produced by the different models for the time period ranging from 1 to 15 March
2017, and for the three large metropolitan areas: Beijing, Shanghai and Guangzhou. In
Figure 8, we compare the median concentrations of the three species with the median
values derived from the different measurements provided by the network of instruments
deployed in the three cities. The median model values are represented by the red curves,
while the shaded areas highlight the dispersion of the calculated concentrations around the
median values.
*Beijing*. Here the predictions of the $PM_{2.5}$ concentrations follow very closely the
observations. Two events with relatively high aerosol loads are visible, the first one between
2 and 5 March and the second one on 11 March. In the case of $NO_2$, the models reproduce





fairly well the daily variability reported by the monitoring stations, but on the average they
slightly underestimate the concentrations values. The high concentrations appearing
between 2 and 5 March and between 10 and 11 March are well captured by the median of
the models. Finally, the models reproduce the diurnal variability in the ozone
concentrations, but they overestimate these concentrations by typically 20 μg m$^{-3}$.
*Shanghai*. The calculated median concentrations of $PM_{2.5}$ are in good agreement with the
observations, especially between 10 and 15 March. During the first part of the simulation,
the mean measured and calculated values are close, but the models fail to capture high
peaks such as those observed on 3, 6, 8 and 9 March. In the case of $NO_2$, the agreement
between calculated and measured concentrations is good. Again, the models severely
overestimate the ozone concentrations.
*Guangzhou.* The median concentration of $PM_{2.5}$ provided by the model is similar to the
observation between 1 and 7 March. However, the model underestimates the
concentrations between 7 and 11 March and overestimates them between 12 and 14
March. For $NO_2$, the agreement between models and measurements is relatively good
during the first days of the month, but the models fail to reproduce the substantial daily
variability observed after 6 March. Ozone is well simulated in this particular urban area,
even though the daily peaks are sometimes over- or underestimated.

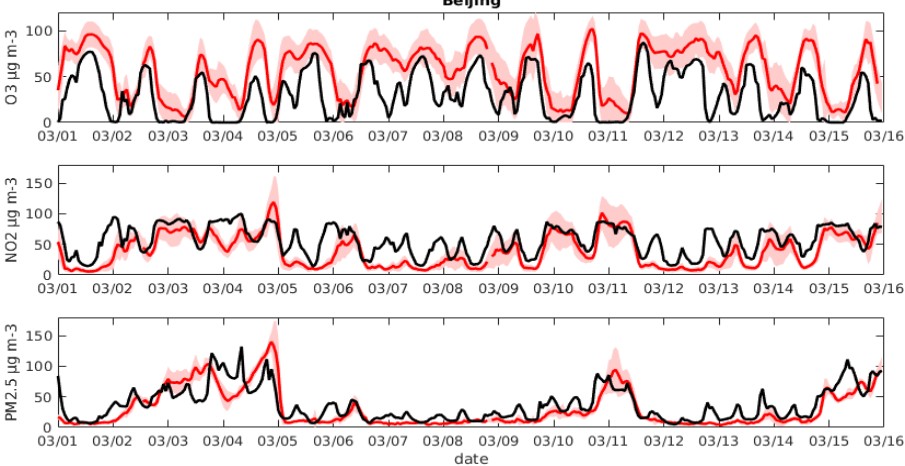






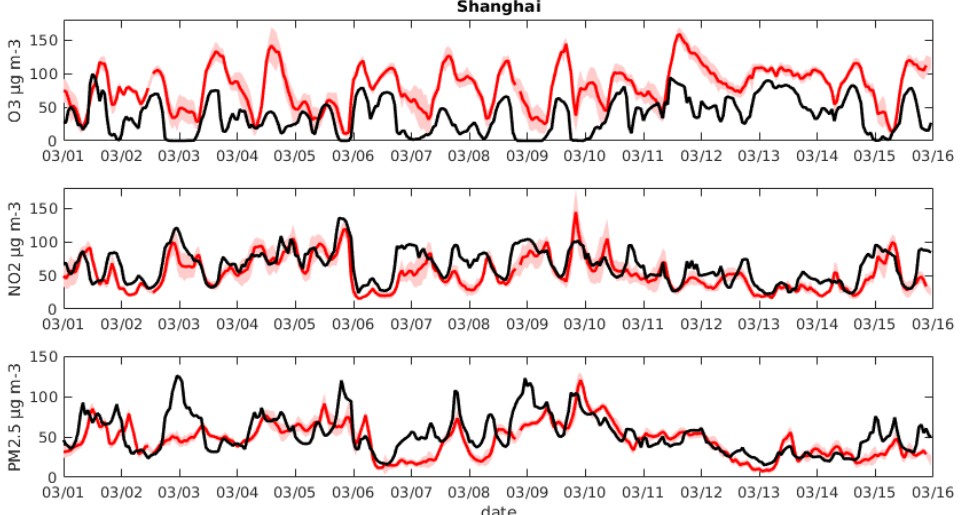


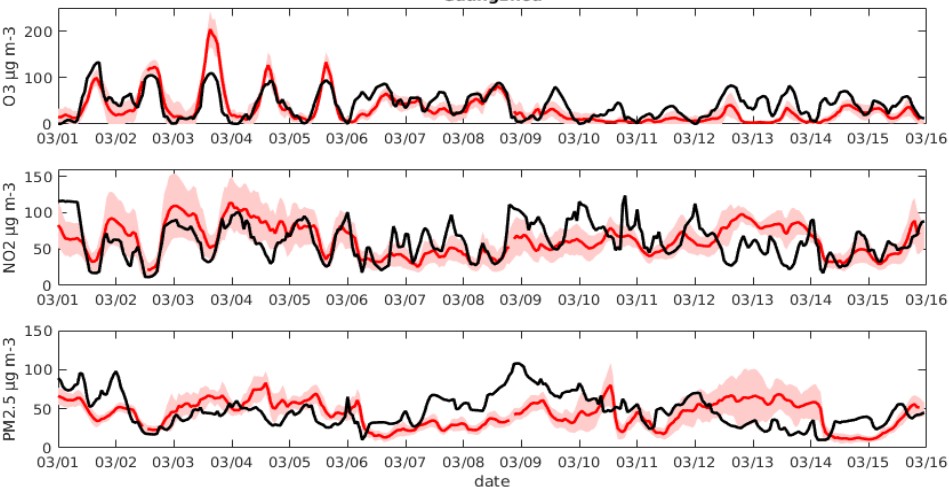


**Figure 8.** Evolution of the surface concentrations of ozone, nitrogen dioxide and particulate matter
(diameter less than 2.5 microns). In red: median of calculated values by the different models, and in
black: observed median concentrations.

***5.3. Statistical Errors***

In order to measure the performance of the individual models involved in the present inter-
comparison, we have calculated statistical measures of the model results for the chosen
period of 1-15 March 2017. These measures include the mean bias (BIAS), the mean
normalized bias (MNMBIAS), the root mean square error (RMSE), the fractional gross error
(FGE) and the correlation coefficient for ozone, NO₂ and PM2.5 (Table 4). They apply to the



data for the 37 cities considered in the MarcoPolo-Panda forecast system. The same
statistical measures are also provided for the ensemble median.
**Table 4: For the period 1st to 15th March 2017, statistical measures (mean bias (BIAS),**
**mean normalized bias (MNB), root mean square error (RMSE), FGE (fractional gross error)**
**and correlation coefficient calculated for the forecast of $O_3$, $NO_2$ and $PM_{2.5}$ concentrations**
**for all models and for the ensemble median at all stations/cities, for which the**
**MarcoPolo-Panda Forecast is available. The correlation is based on 1-hourly data.**

|  |  | Ensemble Median | CHIMERE | IFS | WRF-Chem SMS | SILAM | WRF-Chem MPIM | EMEP | LOTOS-EUROS |
|---|---|---|---|---|---|---|---|---|---|
| BIAS ($\mu g\ m^{-3}$) | O3 | -14.7 | -5.9 | -13.1 | 13.2 | -25.8 | -23.9 | -23.3 | -4.0 |
|  | NO2 | -3.0 | -4.8 | -2.0 | -4.2 | -3.1 | 8.4 | 11.2 | -20.7 |
|  | PM2.5 | 3.7 | -2.0 | 39.7 | -4.5 | 21.7 | 5.5 | 12.4 | -4.7 |
| MNB (%) | O3 | -41% | -24% | -51% | 13% | -74% | -69% | -74% | -7% |
|  | NO2 | -8% | -18% | -13% | -19% | -11% | 13% | 15% | -52% |
|  | PM2.5 | 8% | -4% | 44% | -18% | 22% | 11% | 9% | -7% |
| RMSE ($\mu g\ m^{-3}$) | O3 | 32.8 | 27.0 | 29.4 | 41.8 | 44.6 | 44.7 | 42.9 | 37.2 |
|  | NO2 | 21.8 | 24.4 | 23.1 | 31.9 | 28.5 | 28.9 | 34.0 | 34.4 |
|  | PM2.5 | 30.2 | 31.5 | 71.3 | 35.8 | 47.7 | 39.1 | 52.4 | 27.3 |
| FGE (%) | O3 | 70% | 58% | 72% | 64% | 99% | 97% | 99% | 65% |
|  | NO2 | 38% | 45% | 44% | 53% | 51% | 43% | 48% | 66% |
|  | PM2.5 | 38% | 44% | 62% | 54% | 52% | 49% | 47% | 39% |
| Corr. Coeff. | O3 | 0.60 | 0.70 | 0.72 | 0.45 | 0.32 | 0.32 | 0.39 | 0.38 |
|  | NO2 | 0.64 | 0.62 | 0.65 | 0.47 | 0.41 | 0.50 | 0.46 | 0.31 |
|  | PM2.5 | 0.62 | 0.55 | 0.47 | 0.54 | 0.66 | 0.36 | 0.49 | 0.64 |

When examining the mean bias of the ensemble median, the values are equal to -14.7, -3.0
and +3.7 $\mu g\ m^{-3}$ for ozone, $NO_2$ and PM2.5, respectively, to be compared to mean
concentration values of the order of 50 $\mu g\ m^{-3}$ for these three different species. Table 4
shows in the case of ozone, individual models are characterized by biases ranging from -25.8
(SILAM) to +13.2 $\mu g\ m^{-3}$ (WRF-Chem-SMS) with the smallest absolute value equal to 5.9 $\mu g$
$m^{-3}$ (CHIMERE) The corresponding numbers range from − 20.7 $\mu g\ m^{-3}$ (LOTOS-EUROS) to +
11.2 $\mu g\ m^{-3}$ (EMEP) with the smallest absolute bias of -2.0 $\mu g\ m^{-3}$ (IFS) for $NO_2$. For PM2.5,
they range from -4.7 $\mu g\ m^{-3}$ (LOTOS-EUROS) to +39.6 $\mu g\ m^{-3}$ (IFS) with the smallest absolute
value equal to -2.0 $\mu g\ m^{-3}$ (CHIMERE). In general, during the period chosen for the inter-
comparison, the models underestimate the ozone and $NO_2$ concentrations and
overestimate the concentration of PM2.5. The table also shows that the RMSE for the
median values for ozone, $NO_2$ and PM2.5 are 32.8, 21.8 and 30.2 $\mu g\ m^{-3}$, respectively. With
some exception (CHIMERE and IFS for ozone, LOTOS-EUROS for PM2.5), these values are



lower than the RMSE derived by individual models. The highest values for RSME are 44.7 μg
m$^{-3}$ (WRF-Chem-MPIM) in the case of ozone, 34.4 (LOTOS EUROS) in the case of NO$_2$, and
71.3 (IFS) in the case of PM2.5. The smallest RMSE are equal to 27.0 μg m$^{-3}$ (CHIMERE) in the
case of ozone, 23.1 μg m$^{-3}$ (IFS) in the case of NO$_2$ and 27.3 μg m$^{-3}$ in the case of PM2.5
(LOTOS-EUROS). The correlation coefficient for the ensemble median is of the order of 0.6
for the three species, which in most cases is higher than the values derived from individual
model forecasts. There are few exceptions, however. The correlation coefficients are higher
in the forecast of ozone by CHIMERE (0.70) and IFS (0.72), in the case of NO$_2$ by IFS (0.65)
and in the case of PM2.5 by SILAM (0.66) and LOTOS-EUROS (0.64). Table 5 summarizes the
models that have achieved the best performance from the point of view of the mean bias,
the RMSE and the correlation coefficient.
.
**Table 5. Best Model Performance**

| Statistical Variable | Best performance ozone | Best performance NO2 | Best performance PM2.5 |
|---|---|---|---|
| Mean Bias | LOTOS-EUROS | IFS | CHIMERE |
| RMSE | CHIMERE | IFS | LOTOS-EUROS |
| Correlation coefficient | IFS | WRF-Chem MPIM | SILAM |

***5.4. Time Evolution of Individual Forecasts***
The time evolution of predicted concentration values at Beijing by 5 different models
involved in the inter-comparison is provided in Figure 9 for the period of 1-15 March 2017.
An examination of the figure shows that, during most days, the daytime height of the PBL
reaches 2500 – 3000 m with an exception on 2 to 5 March, when the height does not
exceed 1000 m. Interestingly, during this period, the observed concentration of particulates,
of NO$_2$ and of SO$_2$, strongly influenced by surface emissions, are significantly higher than
during the following days. During the same days, the night-time concentration of ozone is
relatively low. On March 10, one also observes high surface concentrations of emitted
species and low concentration of night-time ozone, even though the calculated PBL height is
not particularly low. One should mention here that in several models (i.e., EMEP, LOTOS-
EUROS) the information on the PBL is deduced from the IFS forecast, while in other models
(such as WRF-Chem-MPIM and WRFChem-SMS) the PBL height is derived independently. In
the case of WRF-Chem-MPI, however, the calculation makes of meteorological data taken
from the IFS model.
In most cases, the models capture relatively well the day-to-day variability in the species
concentrations. The agreement with observations is generally good in the case of PM2.5 and
PM10, except in the case of the IFS model, which considerably overestimates the
concentrations, mainly because of a regional overestimation of the OM emissions and a lack
of a diurnal variation of the emission The anthropogenic OM emissions in IFS are
parameterised based on anthropogenic CO emissions following Spracklen et al. (2017). The
relatively high CO emission in this region may require a reduced conversion factor between
OM and CO emissions. The main contribution to PM overestimation of IFS came from the





night-time values (see next Section). Since night-time overestimation also occurs for $NO_2$, a
lack of vertical mixing during the night in IFS could cause the night time overestimation of
the surface values. As already noted, the models tend to underestimate the ozone
concentrations, perhaps due to a slight overestimation of the nitrogen oxide concentrations.
Another possible explanation is an underestimation of the VOC sources. Routine
measurements of VOCs, however, are not available. The need for such measurements,
however, needs to be stressed.
The model comparison reported here also shows differences between models in the case of
NO, which should probably be attributed to differences in the emissions and emission
injection heights of this species and in the formulation of vertical mixing in the boundary
layer. Here again, measurements of NO in addition to those of $NO_2$ and ozone would be
useful. Finally, one notes in Figure 9 is the relatively good agreement between models (with
the exception of the IFS model) regarding the time evolution of odd oxygen ($Ox = O_3 + NO_2$).
The models, however, slightly underestimate the absolute values of the Ox concentration.

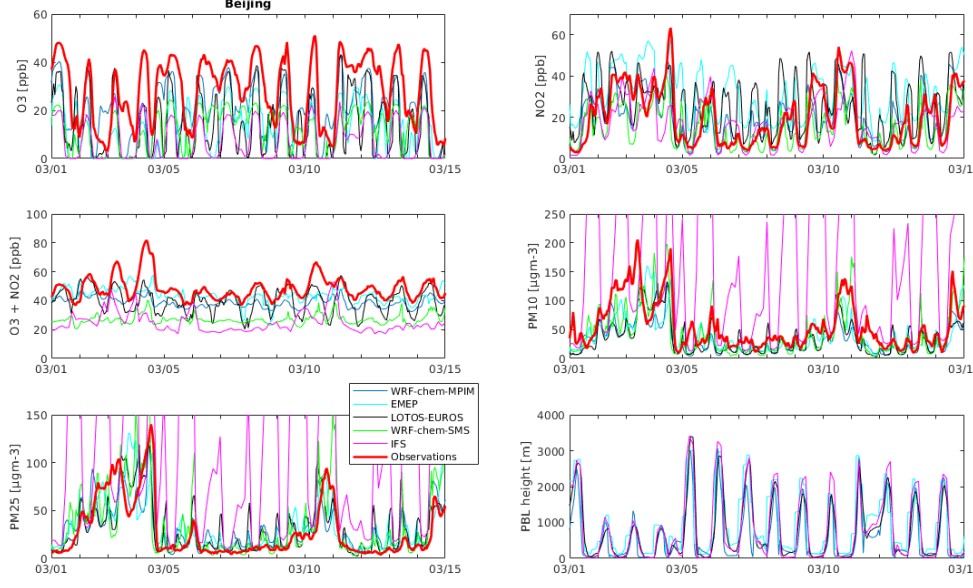

**Figure 9.** Forecast of the chemical concentrations of ozone, $NO_2$, PM2.5, and PM10 at Beijing
between 1 and 15 March 2017 by the different models involved in the inter-comparison conducted
in the present study. The calculated values of $O_X = O_3 + NO_2$ as well as the height of the planetary
boundary layer (PBL) are also shown. The mean values from the measurements made at the
different monitoring stations of Beijing are shown by the thick red line.
***5.5. Diurnal Variations***
In order to evaluate the behaviour of the different models regarding their ability to
reproduce the diurnal variation in the surface concentrations of ozone, $NO_2$ and PM2.5, we





have calculated the mean diurnal variations over the period of 1-15 March 2017 averaged
for the 34 cities included in our analysis (3 of the 37 cities, located in the western part of the
country, and adopted in the MarcoPolo-Panda prediction system have not been considered
in this analysis). The resulting results are shown in Figure 10 for ozone and $NO_2$ (expressed
in $\mu g\ m^{-3}$). We have added the corresponding diurnal evolution of Ox (expressed in ppbv)
defined as the sum of the ozone and $NO_2$ mixing ratios. This last chemical variable has the
advantage that it is not affected by the fast interchange (null cycle) between ozone and $NO_2$
by the reactions $NO + O_3$, $NO_2 + hv$ and $O + O_2 + M$. Since this cycle tends to transfer "odd
oxygen" from ozone to $NO_2$ after sunset and from $NO_2$ to ozone after sunrise, the Ox
variable is less variable than its two components $NO_2$ and $O_3$ over a diurnal cycle. Figure 10
shows that, when averaging over the 34 largest Chinese cities, the diurnal variation of the
ensemble median is in good agreement with the observation in the case of $NO_2$. In the case
of ozone, the median values are somewhat underestimated in late morning and in the
afternoon. A similar situation is found in the case of Ox. The RMSE for ozone and $NO_2$, also
shown on the figure, is generally lower in the case of the ensemble median than for the
individual models. In the case of PM2.5, however, the RMSE of two models, CHIMERE and
IFS are smaller than the RMSE of the ensemble median (not shown here). The mean bias of
the ensemble median for $NO_2$ and ozone is generally smaller than that of the individual
models. In the case of Ox, some models exhibit a positive bias (WRF-Chem SMS), while
others (e.g. SILAM) are characterized by a negative bias.
Figures 11. a, b, c show similar estimates of the diurnal variation in the three large cities of
China: Beijing, Shanghai and Guangzhou.  These graphs show that the ozone forecast from
the ensemble median is lower than observed values during the entire day both in Beijing
and in Shanghai. In Guangzhou, however, ozone is slightly overestimated by the prediction.
In the case of $NO_2$, the surface concentrations are overestimated in Beijing and to a lesser
extent in Shanghai, with the largest over-prediction occurring during night-time, when the
planetary boundary layer is very thin and vertical mixing almost shut off. At the same time,
ozone is negatively biased due to its efficient titration by $NO_x$. In the three cities, the RMSE
of $NO_2$, ozone and Ox appear to be largest at sunset. Thus, a general issue with the
MarcoPolo-Panda prediction system is the overestimation of surface $NO_2$ and the
underestimation of ozone concentrations during night-time.
In the case of PM2.5, one of the models involved (IFS) strongly overestimates the
concentrations during night-time, but is in fair agreement with observations during daytime.
This issue may again reflect a problem with the formulation of species dispersion in the
planetary boundary layer. It may also be due to the lack of specified diurnal variation in the
emission of primary pollutants as well as to the increased night-time stability.



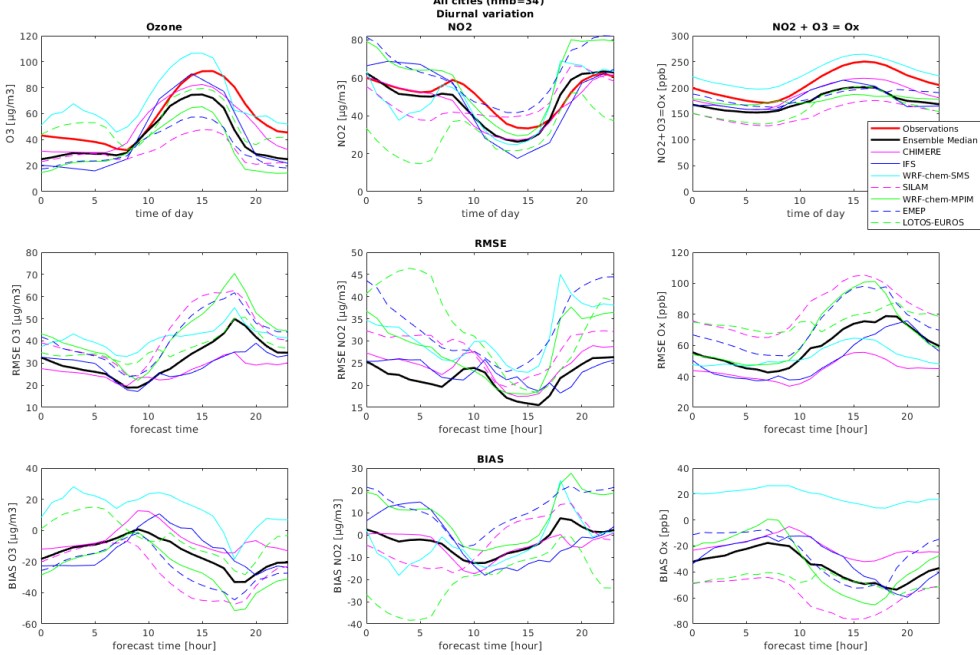

**Figure 10.** Upper panel: Diurnal variation of ozone (left), $NO_2$ (middle) and $Ox = NO_2 + O_3$ (right) for the period $1^{st}$ - $15^{th}$ March 2017 for all cities included in the MarcoPolo-Panda Prediction system for all seven models and the ensemble median, and the observations (red line). Middle panel: Root Mean Square Error (RMSE) for ozone (left), $NO_2$ (middle) and Ox (right). Lower panel: Bias for ozone (left), $NO_2$ (middle) and Ox (right) for all models and for the ensemble median (black line).





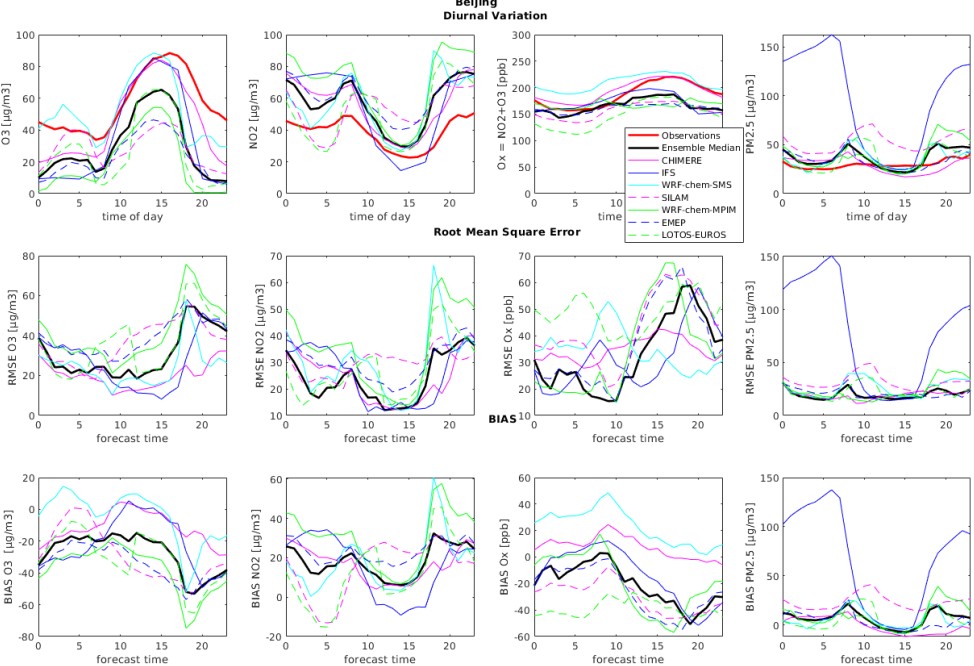

**Figure 11.a.** Same as Figure 10, but for the urban area of Beijing. The statistical variables for PM2.5
are also included.





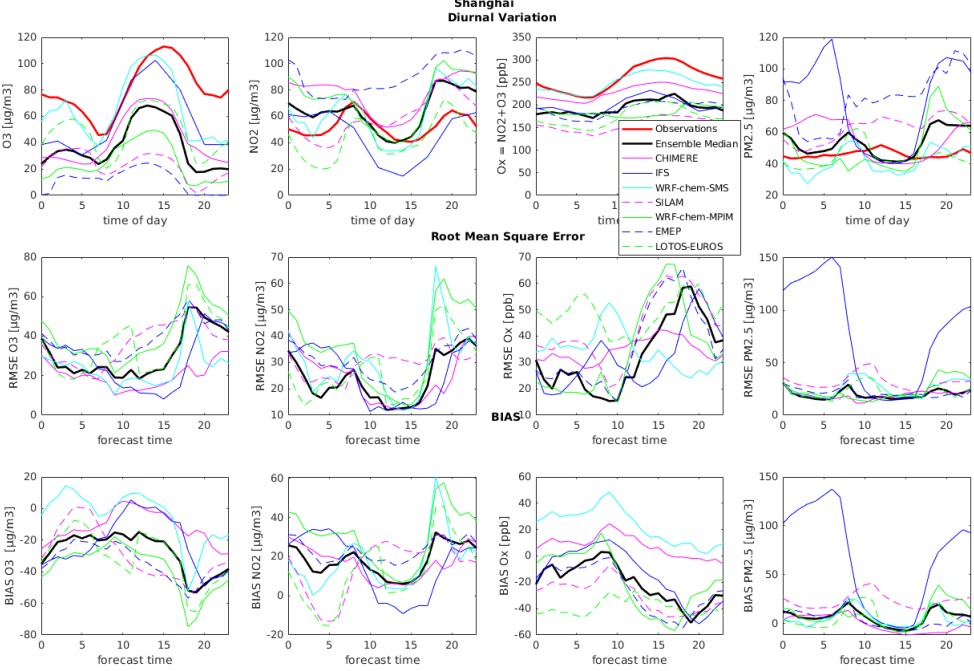

**Figure 11b.** Same as Figure 10, but for the urban area of Shanghai. The statistical variables for PM2.5
are also included.





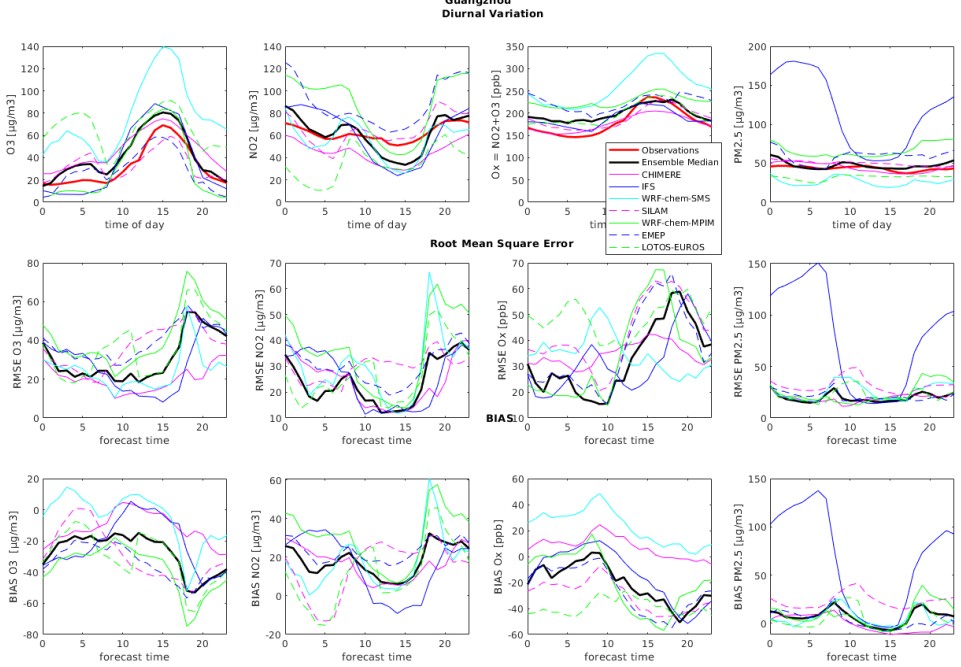

**Figure 11c.** Same as Figure 10, but for the urban area of Guangzhou. The statistical variables for
PM2.5 are also included.

**6. Approaches to Improve the Forecasts**
The inter-comparison presented in the previous sections provides useful information and
represents the basis on which the accuracy of the model predictions can be improved. Since
the models have been developed rather independently and the choices about input
parameters such as emissions, chemical schemes and adopted weather forecasts have been
based on best judgement by these individual teams, a statistical treatment of the model
results (e.g., determination of averages and standard deviation) provides in general more
reliable information than the data provided by the individual model components of the
ensemble. The examination of the model output reveals, however, some systematic biases
that could be reduced by identifying the likely cause of these errors.
A simple approach is to recognize that the failure of models to correctly predict air quality
could result from several factors: (1) errors in the adopted emissions and the formulation of
boundary layer dispersion best diagnosed by analysing the ability of the model to reproduce
the monthly mean surface concentrations of chemical species; (2) errors or omission in the
adopted chemical scheme leading to inaccuracies in the calculated mean diurnal variations
in the concentrations of secondary species; and (3) inaccuracies in the adopted weather
forecasts leading to poorly calculated day-to-day variations in the calculated chemical fields.
In this later case, one should distinguish between fundamental model biases (i.e., the
representation of PBL mixing, a bias that is intrinsic to the models) and the increasing error



in the forecast of synoptic weather patterns as the model integration proceeds. This
probably provides an oversimplified view of the causes of errors in chemical weather
forecasts, but it offers a simple approach to address some issues in the models and hence to
improve the predictions.
A first step towards the improvement of the different model components will be to conduct
additional simulations by adopting the same best available emissions data and the same
meteorological forecasts. Remaining differences between the models will be due in large
part (although not exclusively) to the adopted chemical scheme and the formulation of
boundary layer processes. An additional step would be to bring the different formulations of
chemistry closer together by at least harmonizing the adopted rate constants and using the
same module to calculate photodissociation rates. Finally, it would be interesting to assess
the differences in chemical weather predictions resulting from the adopted meteorological
forecasts. In particular, it would be important to better constraint the differences in the
photolysis rates resulting from the adopted or calculated concentrations of aerosols and in
cloudiness. One single model could be run for several days with the weather predictions
produced by different meteorological centres.
Finally, a few specific issues from the present inter-comparison require attention:
(1) Most models overestimate the surface levels of $NO_2$ and PM2.5 as well as other
species emitted at the surface, specifically during night-time. The largest
discrepancies appear around 18"00 LT when the surface cools and the boundary
layer collapses and the emitted species remain trapped in the lowest model layers.
Evidently, these models underestimate the vertical exchanges between layers
probably produced by the turbulence thermally or mechanically generated by the
presence of buildings. Such effects are not accounted for in models that do include a
specialized urban formulation. The overestimation of $NO_2$ during night-time leads to
the titration of ozone near the surface and hence an underestimation of the
concentration of this gas. The emission injection height is also a relevant factor here,
which can largely influence results. During night-time emissions from stacks may be
emitted above the mixing layer. However if the injection height in the model is put at
lower altitude (or even at the surface) this could lead to overestimation of emissions.
The LOTOS-EUROS model evaluated the impact of emission injection heights. An
update of the emission heights was tested that injects emissions from industry at
lower heights, representing that the number of high stacks is limited (not that
contrarily to most models, in the case of LOTOS-EUROS the concentrations at night-
time are often underestimated (see Figures 10 and 11). Figure 12 shows diurnal
cycles of the simulated PM2.5 concentrations in the city of Chengdu, averaged over
an entire year. The updated emission heights clearly have a large (positive) impact
on the simulations.
(2) Daytime concentrations of ozone are generally underestimated in most regions of
eastern China, even when the level of $NO_2$ is in reasonable agreement with the
values reported by the monitoring stations. The discrepancy could be caused by an
underestimation of the emissions of some VOCs, especially in urban areas where
ozone is often VOC-limited. More work is required to investigate this question.



(3) Emissions of primarily pollutants are changing extremely rapidly in China. The adopted emissions inventories usually reflect to the situation a few years before present-day. Since the current emissions have decreased significantly in some urban areas of China in response to measures taken by the authorities, the emissions used in this case for current forecasts may be overestimated. For example, the EMEP model team applied a reduction in NOx emissions after the study period of March 2017 and thereby, through less ozone titration, reduced the severe underestimation of ozone.

(4) Land-use data. Due to the rapid development occurring in particular in the Eastern part of China, land-use data and vegetation change rapidly, and data sets in the model may not accurately reflect the current situation. This has an influence on emissions (including biogenic) but also deposition of pollutants and even meteorology. Land-use data should be updated using satellite observations, urban planning maps and other data sources.

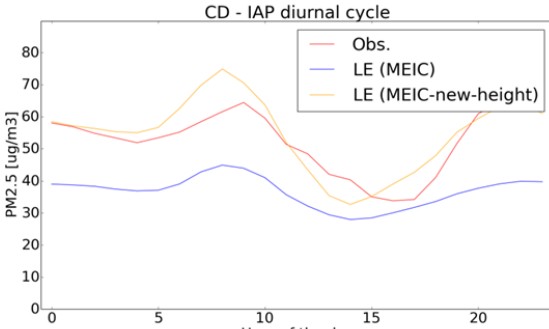

**Figure 12.** Annually averaged diurnal evolution of the PM2.5 concentrations in the city of Chengdu simulated for different values of the particulate injection height.

## 7. Conclusions

An operational multi-model air quality forecast system has been established through a close cooperation between European and Chinese research groups and with the support of the European Commission (7[th] Framework Programme). This system provides daily forecasts for the surface concentration of key pollutants in eastern China, and particularly in the major urban centres of the country. These predictions are posted on a dedicated website (www.marcopolo-panda.eu), where they are compared hour by hour to surface measurements for each city, performed at the monitoring stations deployed in China by the PM2.5 network (www.pm25.int).

The discussions presented in this paper show that in most cases, the model ensemble reproduces quite satisfactorily the synoptic behaviour and the day-to-day variability of the concentrations of ozone and particulate matter and, in particular, predicts the development of most air pollution episodes a few days before their occurrence. This must be attributed to the quality of the weather forecasts at the synoptic scales that are used for the calculation



of chemical species. Overall and in spite of some discrepancies that have been highlighted in
the previous sections, the forecast system can therefore be regarded as successful.
The system is in its early phase of development and the purpose of the inter-comparison
exercise presented here was to diagnose differences between models and perhaps identify
errors. An important objective was to determine ways by which the models could be
improved. Even though, in many instances, the surface concentrations are in good or fair
agreement with the measured values, differences between calculated and observed values
can occasionally be substantial. These occasional differences are often attributed to
inaccuracies in the weather forecasts for specific days, but errors in the adopted surface
emissions and PBL exchanges or the simplifications introduced in the adopted chemical and
aerosol schemes can also be substantial.
The degree by which the concentrations derived by global and regional models, even at high
spatial resolution, can be compared with local measurements made in a complex urban
canopy remains an important issue that requires further investigation. The insertion of
more detailed land-use modules or of a large eddy simulation system in the chemical
transport models should be considered in future studies.

## Data Availability

The models described here are used operationally by the participating research and service organizations
involved in the present study. The data produced by the multi-model forecasting system are available from the
Royal Dutch Meteorological Institute (KNMI). The source codes of the modeling systems can be downloaded
from the following websites: WRF-Chem: http://www2.mmm.ucar.edu/wrf/users/downloads.html, Chimere:
http://www.lmd.polytechnique.fr/chimere/, Silam: http://silam.fmi.fi/, LOTOS-EUROS: https://lotos-
euros.tno.nl/, EMEP: http://www.emep.int/, WRF-CMAQ: https://www.epa.gov/cmaq/cmaq-models-0,
CMAQ: https://github.com/USEPA/CMAQ/

## Acknowledgements


The model inter-comparison presented in the present study has been conducted during a
workshop organized in May 2017 by the Shanghai Meteorological Service (SMS) in China.
The authors thank Jianming Xu for hosting this meeting and providing support to the
participants. The ensemble of models described here has been produced under the Panda
and MarcoPolo projects supported by the European Commission within the Framework
Program 7 (FP7) under grant agreements n°606719 and n°606953. The National Center for
Atmospheric Research (NCAR) is sponsored by the US National Science Foundation.





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
