# Peer review of "Ensemble Forecasts of Air Quality in Eastern China Part 1. Model Description and Implementation of the MarcoPolo-Panda Prediction System, Version 1."

_Geoscientific Model Development, 2018_

## Short Comment (SC1) · 19 Jul 2018

Dear authors,

in my role as Executive editor of GMD, I would like to bring to your attention our Editorial version 1.1:

http://www.geosci-model-dev.net/8/3487/2015/gmd-8-3487-2015.html

This highlights some requirements of papers published in GMD, which is also available

on the GMD website in the 'Manuscript Types' section:

http://www.geoscientific-model-development.net/submission/manuscript_types.html

In particular, please note that for your paper, the following requirement has not been met in the Discussions paper:

- "The main paper must give the model name and version number (or other unique identifier) in the title."

Please add a version number for the MarcoPolo-Panda Prediction System in the title upon your revised submission to GMD.

Yours,

Astrid Kerkweg

---

## Referee Comment (RC1) · Anonymous Referee #1 · 20 Jul 2018

This paper describes an interesting approach to develop ensemble air quality forecasting systems. Specifically, it describes the outcome of 2 EU projects that produced an ensemble forecast system of opportunity. This approach takes existing models developed and applied elsewhere and brought together under this project to produce a forecasting system. Each model member was free to use their inputs of choice (emissions etc.,), This approach is described here and demonstrates how this can be done in a rather direct manner to produce a system. This approach in principal is reproducible in any region as the models can be applied anywhere and global emission estimates and meteorological inputs are available globally. This paper is a straight-forward description of the modeling system. It includes the necessary descriptions of the individual models, inputs used, and how the ensembles were generated. The paper then presents a series of illustrative results focused on model predictions and performance for March 2017. These results are interesting and informative – showing cases where the ensemble performs well and cases where the models and ensemble have difficulty. Other results show some insights into why the models behave differently. Quantitative statistical results are presented on for a 2-week period of March. The limitation of the paper is that the results presented are only illustrative and for a single month. The paper is Part 1 implying that there will be additional papers submitted. It would be helpful to state in the paper what follow-on papers are being considered. This will help clarify the scope of this part 1 paper. Presumably a more rigorous evaluation will be presented in another paper based on at least an entire year. The paper includes discussion of limitations and potential ways to improve the forecasting system, which is informative. In summary this paper presents an interesting new prediction system for which the approach is applicable in other regions. This warrants publishing. I offer a few comments below for consideration. The references are pretty Europe centric. Section starting on line 137 could consider adding an earlier reference to ensembles McKeen, S. et al., Assessment of an ensemble of seven real-time ozone forecasts over Eastern North America during the summer of 2004, J. Geophys. Res., 110, D21307, doi: 10.1029/2005JD008888, 2005 I suggest adding a separate section on model evaluation. What systematic evaluations are performed, what data is used, how the data was made available, are their plans in the system for retrospective evaluation (say every year using additional data (not available in real time) and finalized data, etc., This is an important component of a forecast system. It would be interesting to present lessons learned – about establishing such a system in another region. What guidance can be given?
* * *
[Figure]

2018.

---

## Referee Comment (RC2) · Anonymous Referee #2 · 15 Sep 2018

General Summary This paper describes a newly developed ensemble forecasting that is being used to produce daily deterministic and probabilistic forecasts of air quality in China. This is a timely study in light of the fact that air quality has become a really serious environmental concern in Asia. Air quality forecasts, such as those described here, enhances the ability of air quality managers to warn the public in advance of the forthcoming air pollution episodes. The ensemble system is described in detail with both the capabilities and shortcomings for a period in March 2017. I think the paper is suitable for publication in GMD but have some minor specific comments that are listed

below.

Specific comments

Line 115: Change "include assimilated data" to "assimilate data".

Line 145: I guess you mean to say "Numerical weather forecasting at seasonal scales..." here.

Line 152: In addition to meteorological forecasts, I think it would be useful to drive a single model with an ensemble of emission scenarios and chemistry.

Line 158, 161 and 162: I suggest naming all the processes instead of leaving the reader with the curiosity of what "...." mean here.

Lines 177-190: I suggest defining all the acronyms (e.g., WRF-Che, WRF-CMAQ, SILAM etc.) upon their first use here.

Line 207: Change "aata" to "data".

Line 214: Suggesting adding NOx to ozone-CO-NMVOC.

Line 237: Could you please provide a brief summary (2-3 lines) of the overall performance of IFS over March-May 2017?

Section 2.2: Please provide information about at what resolution CHIMERE forecasts were produced.

Line 275: Change to Fast et al., 2006.

Line 319: Spell out STEAM.

Line 433: Could you say more about how anthropogenic emissions are adjusted every week? Do you employ a machine-learning approach?

Line 440: I guess you mean "ideal" profiles and not "idea".

Line 453: All these papers focus on the U.S. It is okay to cite these papers but it would

be useful to add few references for applications of CMAQ over China.

Section 5.1: Can you say something about the role of representativeness errors in model-observation discrepancies? Will the model performance change if you isolate the comparison only to rural sites?

Line 754: Is better performance of IFS related to assimilation?

Lines 735-758 are the same as 760-784. Please remove the duplication.

Lines 811- 812: It is well known that models have difficulties in reproducing nighttime concentrations of air pollutants including ozone. How does the model perform for day-time ozone? Section 5.2 provides some information about the daytime performance in three metro areas but it will be good to examine and discuss spatial patterns of daytime ozone in particular.

Line 906: Change RSME to RMSE.

Figs. 8 and 9: I am somewhat puzzled by the PM2.5 panels in Figs 8 and 9. For Beijing, ensemble median (Fig. 8) is lower than the observations for March 5-10 while all models show higher PM2.5 values than the observations in Fig. 9. I also suggest using the same color for observations throughout. Fig. 8 shows observations in black and Fig. 9 shows in red. Adding legends to Fig. 8 will also be useful. I was also expecting the spread will be higher in Fig. 8 because IFS has such large value of PM2.5. Similarly, all the models are lower than observations for ozone (Fig. 9) but the median of the models in Fig. 8 is higher than the observed ozone. Please check the plots carefully and revise the discussion.

Line 935: Do you want to say that WRF-Chem-MPI meteorological simulations are driven by IFS?

Line 958: Even the WRF-Chem-SPS does not agree with other models for odd-oxygen.

[Figure]

2018.

---

## Author Comment (AC1) · 8 Oct 2018

Dear Executive Editor,

Thank you for your comment. We have added in the title of the paper the words "Version 1"
* * *

---

## Author Comment (AC2) · 8 Oct 2018

Dear Reviewer,

Thank you very much for your comments and your supportive review of our manuscript. Below are the responses to your specific comments:

The references are pretty Europe centric. Section starting on line 137 could consider adding an earlier reference to ensembles McKeen, S. et al., Assessment of an ensemble of seven real-time ozone forecasts over Eastern North America during the summer

[Figure]

of 2004, J. Geophys. Res., 110, D21307, doi: 10.1029/2005JD008888, 2005

Response: We have added this important reference.

I suggest adding a separate section on model evaluation. What systematic evaluations are performed, what data is used, how the data was made available, are their plans in the system for retrospective evaluation (say every year using additional data (not available in real time) and finalized data, etc., This is an important component of a forecast system. It would be interesting to present lessons learned – about establishing such a system in another region. What guidance can be given?

Response: Another paper with a similar title, but labelled "Part 2: Evaluation of the MarcoPolo Prediction System, Version 1" has been submitted to GMD and is currently under review in the GMD-Discussion. We feel therefore that we should not add a separate section in Part 1, but that we should address the remarks and follow the suggestions made here in the Part 2 paper. The suggestion to discuss Lessons learned about establishing such a system in another region is particularly interesting and will be included in the conclusion of the second paper.

---

## Author Comment (AC3) · 11 Oct 2018

We would like to thank reviewer 2 for the detailed reading of our paper and the suggested changes. Reviewer 2 also detected an error in the paper, which we were able to correct (see below).

Line 115: Change "include assimilated data" to "assimilate data". Response: Done

Line 145: I guess you mean to say "Numerical weather forecasting at seasonal scales. . ." here. Response: Our point concerns numerical weather forecast in general and not

only on seasonal scales.

Line 152: In addition to meteorological forecasts, I think it would be useful to drive a single model with an ensemble of emission scenarios and chemistry. Response: We added the words "emission scenarios and chemistry" as suggested by the reviewer.

Line 158, 161 and 162: I suggest naming all the processes instead of leaving the reader with the curiosity of what ". . .." mean here. Response: we removed the signs "..." and used "e.g." instead.

Lines 177-190: I suggest defining all the acronyms (e.g., WRF-Che, WRF-CMAQ, SILAM etc.) upon their first use here. Response: We spelled out all the acronyms except in the case of Chimere, which is the name of the model but is not an acronym.

Line 207: Change "aata" to "data". Response: Done.

Line 214: Suggesting adding NOx to ozone-CO-NMVOC. Response: Done.

Line 237: Could you please provide a brief summary (2-3 lines) of the overall performance of IFS over March-May 2017? Response: It is impossible to provide such an evaluation in a few lines. We prefer to refer the reader to the reports that are available from ECMWF.

Section 2.2: Please provide information about at what resolution CHIMERE forecasts were produced. Response: Done. It is 0.25 degrees.

Line 275: Change to Fast et al., 2006. Line 319: Spell out STEAM. Response: Done.

Line 433: Could you say more about how anthropogenic emissions are adjusted every week? Do you employ a machine-learning approach? Response: The emissions for several species such as SO2, CO, PM2.5, PM10, etc. are adjusted by applying a factor that accounts for error in the predicted concentration the week before. See more details below. The text has been slightly adjusted. No machine-learning.

Line 440: I guess you mean "ideal" profiles and not "idea". Response: Corrected.

Line 453: All these papers focus on the U.S. It is okay to cite these papers but it would C2 GMDD be useful to add few references for applications of CMAQ over China. Response: We added two references and some text.

Section 5.1: Can you say something about the role of representativeness errors in model-observation discrepancies? Will the model performance change if you isolate the comparison only to rural sites? Response: This is discussed in Paper 2 by Petersen et al. (submitted to GMD). A sentence mentioning the representativeness error and referring to Paper 2 has been added in the introduction of the paper.

Line 754: Is better performance of IFS related to assimilation? Response: yes. Text added.

Lines 735-758 are the same as 760-784. Please remove the duplication. Thanks for noting this. Duplicated text is removed.

Lines 811- 812: It is well known that models have difficulties in reproducing nighttime concentrations of air pollutants including ozone. How does the model perform for daytime ozone? Section 5.2 provides some information about the daytime performance in three metro areas but it will be good to examine and discuss spatial patterns of daytime ozone in particular. Response: This is discussed in detail in Paper Part 2 by Petersen et al. (Section 5.5).

Line 906: Change RSME to RMSE. Response: Done.

Figs. 8 and 9: I am somewhat puzzled by the PM2.5 panels in Figs 8 and 9. For Beijing, ensemble median (Fig. 8) is lower than the observations for March 5-10 while all models show higher PM2.5 values than the observations in Fig. 9. I also suggest using the same color for observations throughout. Fig. 8 shows observations in black and Fig. 9 shows in red. Adding legends to Fig. 8 will also be useful. I was also expecting the spread will be higher in Fig. 8 because IFS has such large value of PM2.5. Similarly, all the models are lower than observations for ozone (Fig. 9) but
the median of the models in Fig. 8 is higher than the observed ozone. Please check the plots carefully and revise the discussion. Line 935: Do you want to say that WRF-Chem-MPI meteorological simulations are driven by IFS? Response: here the model has detected an error in the paper (thank you!). In fact, in all figures including Figure 8, the calculated median values are in black and the observations are in red (and not the opposite). When this correction is made, there is no inconsistencies anymore between Figures 8 and 9. The text and the captions have been changed to correct this mistake.

Line 958: Even the WRF-Chem-SPS does not agree with other models for odd-oxygen. Response: We have added that the WRF-Chem-SMS model does not agree with other models for odd-oxygen.

Note regarding the adjustment of the emissions by the Nanjing air quality model.

It should be noted that the anthropogenic emissions in mainland China are not fixed in this system, but are automatically adjusted every week according to the system performances in the past week. Briefly, there are 334 prefectural-level divisions in mainland China, and in each prefectural-level division, the mean relative deviations of SO2 ($\Delta$SO2), NO2 ($\Delta$NO2), CO ($\Delta$CO), PM10 ($\Delta$pm10) and PM2.5($\Delta$pm2.5) between the predicted and observed concentrations for the past week are calculated every Sunday. In each division, the spatial distributions of each pollutant emission are assumed to be right, but the emission levels have deviations. A series of scaling factors are given to adjust the emissions of SO2 (ðİŽŇSO2), NOx (ðİŽŇNOx), CO (ðİŽŇCO), PM2.5 (ðİŽŇpm2.5), and PM10 (ðİŽŇpm10), respectively, namely, the emission after adjustment is equal to the original emission multiply the scaling factor. Meanwhile, We also assume that the relationships between the concentrations and the emissions are linear, and the bias of NO2 in each division is all caused by local NOx emission, while the ones of SO2, CO, PM2.5 and PM10 are 60% contributed by local emission errors, and the rests are transported from the other divisions. So, for NOx, ðİŽŇNOx = ðİŽŇNOx, old *1/(1+$\Delta$NO2); for SO2 and CO, ðİŽŇ = ðİŽŇold *(0.4+0.6/(1+$\Delta$)); for PM2.5, we assume that 50% of PM2.5 is from the primary PM2.5 emissions, therefore,

[Figure]

ðİŽŇPM2.5 = ðİŽŇPM2.5, old *(0.7+0.3/(1+$\Delta$PM2.5)); for PM10, the PM10 emission in the inventory only includes coarse particle, therefore, the predicted and observed coarse particle concentration (i.e., PM10 minus PM2.5) are used during the calculation of $\Delta$pm10, thus, ðİŽŇPM10 = ðİŽŇPM10, old *(0.4+0.6/(1+$\Delta$PM10)). The ðİŽŇold represents the scaling factor of last week. It is noted that the emission of NH3 is not adjusted and each VOC species has the same scaling factor, and is equal to the one of NOx.
* * *